# The transcription factor AP2XI-2 is a key negative regulator of *Toxoplasma gondii* merogony

Jin-Lei Wang [1] ✉, Ting-Ting Li[1], Nian-Zhang Zhang[1], Meng Wang[1], Li-Xiu Sun[1], Zhi-Wei Zhang[1], Bao-Quan Fu[1], Hany M. Elsheikha [2] ✉ & Xing-Quan Zhu [3] ✉

Sexual development in *Toxoplasma gondii* is a multistep process that culminates in the production of oocysts, constituting approximately 50% of human infections. However, the molecular mechanisms governing sexual commitment in this parasite remain poorly understood. Here, we demonstrate that the transcription factors AP2XI-2 and AP2XII-1 act as negative regulators, suppressing merozoite-primed pre-sexual commitment during asexual development. Depletion of AP2XI-2 in type II Pru strain induces merogony and production of mature merozoites in an alkaline medium but not in a neutral medium. In contrast, AP2XII-1-depleted Pru strain undergoes several rounds of merogony and produces merozoites in a neutral medium, with more pronounced effects observed under alkaline conditions. Additionally, we identified two additional AP2XI-2-interacting proteins involved in repressing merozoite programming. These findings underscore the intricate regulation of pre-sexual commitment by a network of factors and suggest that AP2XI-2 or AP2XII-1-depleted Pru parasites can serve as a model for studying merogony in vitro.

*Toxoplasma gondii* is a ubiquitous human and veterinary protozoan parasite with a multistage and heteroxenous life cycle involving sexual reproduction strictly in the gut mucosa of the definitive felid host and asexual reproduction in the intermediate host including many warm-blooded animals and humans[1]. Although primary *T. gondii* infection by rapidly replicating tachyzoites can cause mild flu-like symptoms in immunocompetent individuals, infection in immunocompromised patients can result in adverse health consequences[2–4]. Likewise, infection in pregnant women can cause miscarriage, stillbirth, or congenital anomalies, including neurological and ocular defects[2–4]. To counter the pressure exerted by the host immune response, tachyzoites differentiate into slowly replicating bradyzoites enclosed within thick-walled tissue cysts, which marks the latent stage of infection. Upon ingestion of the tissue cysts, bradyzoites released from the ingested cysts undergo merogony to produce merozoites, which differentiate into gametes and ultimately form oocysts that are excreted in the feline feces[1].

The ability of *T. gondii* to respond to intrinsic and extrinsic cues is contingent upon complex regulatory systems involving transcriptional regulators. A repertoire of transcriptional factors has been shown to influence gene expression and genetic reprogramming, which mediate parasite infectivity, stage differentiation, persistence, and transmission[5–10]. The family of plant-related transcription factors Apetala−2 (AP2) plays a pivotal role in regulating apicomplexan developmental stage transition[11]. Many members of the apicomplexan AP2 (ApiAP2) family are expressed in a cell cycle- or stage-specific

[1]State Key Laboratory for Animal Disease Control and Prevention, Key Laboratory of Veterinary Parasitology of Gansu Province, Lanzhou Veterinary Research Institute, Chinese Academy of Agricultural Sciences, Lanzhou, Gansu Province 730046, People's Republic of China. [2]Faculty of Medicine and Health Sciences, School of Veterinary Medicine and Science, University of Nottingham, Sutton Bonington Campus, Loughborough LE12 5RD, UK. [3]Laboratory of Parasitic Diseases, College of Veterinary Medicine, Shanxi Agricultural University, Taigu, Shanxi Province 030801, People's Republic of China. ✉e-mail: hany.elsheikha@nottingham.ac.uk; xingquanzhu1@hotmail.com; wangjinlei90@126.com

manner[12]. Some AP2s coordinate transcriptional regulatory activities by directly or indirectly interacting with epigenetic factors, such as histone acetyltransferase (GCN5) and microrchidia (MORC) transcriptional repressor protein[13,14]. At least 12 AP2-related factors are associated with MORC, which recruits histone deacetylase HDAC3 to control *T. gondii* sexual development[14,15].

Previous studies showed that ApiAP2 factors can function as activators or repressors of gene expression in a parasite stage-specific manner. For instance, AP2IV-3 and AP2XI-4 act as transcriptional activators, whereas AP2IX-4, AP2IX-9, and AP2IV-4 serve as transcriptional repressors to regulate the expression of bradyzoite genes during tachyzoite differentiation into bradyzoites[12,16–19]. The disruption of individual TgAP2 factors partly affects but does not completely block stage differentiation. Despite this knowledge, our understanding of the role of the ApiAP2 transcription factors in the regulatory networks controlling sexual reproduction in *T. gondii* remains limited. Elucidating the mechanisms by which *T. gondii* differentiates into merozoites is a prerequisite for understanding the sexual development in this parasite.

Here, we show that the merozoite-primed pre-sexual commitment in *T. gondii* is under negative transcriptional control by AP2 transcription factors. Specifically, we characterize two AP2 transcription factors, AP2XI-2 and AP2XII-1, which are constitutively expressed in both tachyzoites and bradyzoites but not in merozoites. Genetic ablation of AP2XI-2 in the type II Pru strain hinders bradyzoite differentiation and triggers the merogony process, leading to the formation of polyploid schizonts and mature merozoites in an alkaline medium, but not in a neutral medium. Likewise, AP2XII-1, identified as an AP2XI-2-interacting protein, also suppresses the expression of merozoite-specific genes. Depletion of AP2XII-1 results in initiation of merogony and production of merozoites. Notably, the merogony process is more prominent in the AP2XI-2-depleted strain compared to the AP2XII-1-depleted strain under alkaline conditions. Moreover, two additional AP2XI-2 interacting proteins (TGME49_209500 and TGME49_275680) are identified as regulators of the merogony process. This study provides a comprehensive understanding of how the transcriptional factors AP2XI-2 and AP2XII-1 orchestrate the regulatory networks that govern merozoite-primed pre-sexual commitment in *T. gondii*.

## Results

### The nuclear factor AP2XI-2 is important for *T. gondii* lytic cycle

The AP2XI-2 gene (TGME49_310900) encodes a 232.85-kDa protein containing 2243 amino acids and two AP2 domains near the C-terminus (Supplementary Fig. 1a). Homologs of the AP2XI-2 protein were identified in several coccidian parasites including *Hammondia hammondi*, *Neospora caninum*, *Besnoitia besnoiti*, *Eimeria tenella*, and *Sarcocystis neurona* (Supplementary Fig. 1b). Given the marked negative phenotype score (−5.07) in a genome-wide fitness screen[20], we employed the mini auxin-inducible degron (mAID) system, previously developed for studying essential genes in *T. gondii*[21], to study the biological function of AP2XI-2. To achieve this, we constructed a conditional knockdown strain Pru::AP2XI-2-mAID-6HA (referred to as AP2XI-2-mAID) by endogenously tagging AP2XI-2 with a C-terminal mAID-6HA in the auxin receptor (TIR1) expressing line (PruΔ*ku80Δhxgprt*::TIR1-3Flag). Immunofluorescence assays showed that AP2XI-2 is a nuclear protein that is constitutively expressed in tachyzoites and bradyzoites triggered by alkaline (pH 8.2) conditions (Fig. 1a). The mAID system enabled rapid depletion of AP2XI-2 by treating the parasites with 3-indoleacetic acid (auxin or IAA) as demonstrated by immunofluorescence (Fig. 1a) and immunoblotting (Supplementary Fig. 1c) analysis.

To determine the role of AP2XI-2 in *T. gondii* growth, we examined the ability of the AP2XI-2-mAID tachyzoites to form plaques in HFF monolayers in the presence or absence of IAA. The parental strain grew normally in the presence of IAA, whereas the growth of the AP2XI-2-

mAID strain was significantly inhibited upon the addition of IAA. Interestingly, AP2XI-2-mAID formed very small plaques compared to their growth in the absence of IAA, suggesting that although AP2XI-2 is critical for tachyzoite growth, AP2XI-2-deficient tachyzoites may remain viable (Fig. 1b, c). To test this assumption, CRISPR-Cas9 mediated homologous gene replacement was used to knock out the AP2XI-2 coding gene in the PruΔ*ku80Δhxgprt* strain, and a stable AP2XI-2 knockout strain was successfully isolated after multiple trials (Supplementary Fig. 1d). A similar growth phenotype was observed in the PruΔ*ap2XI-2* strain (Fig. 1b, c).

To determine which step of the parasite lytic cycle was affected by AP2XI-2 deletion, we conducted assays to investigate the parasite invasion, intracellular replication, and egress by using the AP2XI-2-mAID strain. For the invasion assay, AP2XI-2-mAID parasites pretreated for 48 h with or without IAA were used to infect the host cells for 30 min, and the parasite invasion efficiency was then assessed. When AP2XI-2 protein was degraded in the presence of IAA, AP2XI-2-mAID strain showed a significant decrease in cell invasion compared with that of the AP2XI-2-expressing parasites (Fig. 1d). Next, we evaluated whether depletion of AP2XI-2 affects the replication of AP2XI-2-mAID parasites in HFFs 30 h after invasion in the presence of IAA. The degradation of AP2XI-2 by addition of IAA caused a significant reduction in the replication efficiency of AP2XI-2-mAID compared to that of AP2XI-2-expressing parasites (Fig. 1e). We further examined the egress of A23187-stimulated parasites from HFF monolayers after treatment with or without IAA for 48-60 h, and the results showed significant reduction in the ability of AP2XI-2-mAID parasites treated with IAA to egress from the host cells (Fig. 1f). To investigate the impact of AP2XI-2 deletion on the parasite virulence, mice were infected by $2 \times 10^4$ or $2 \times 10^5$ of the PruΔ*ap2XI-2* or parental Pru strain, and their survival was monitored for 30 days. As expected, all mice infected by PruΔ*ap2XI-2*, as confirmed by serological testing, survived without any obvious clinical signs and no cysts were detected in their brains (Fig. 1g, h). In contrast, only 30% mice infected by $2 \times 10^4$ of the parental strain survived and their brain contained a small number of cysts ($48 \pm 22$). These results showed that AP2XI-2 is important for *T. gondii* growth in vitro and in vivo.

### Loss of AP2XI-2 causes morphological defects triggered by alkaline stress

To investigate the function of AP2XI-2 in bradyzoites, the frequency of tachyzoite-to-bradyzoite differentiation in the AP2XI-2-mAID parasites incubated under alkaline growth conditions with or without IAA, was examined by using FITC-*Dolichos biflorus* lectin (DBL) staining, which binds to N-acetylgalactosamine in the parasite cyst wall[22]. After 3 days in an alkaline condition without IAA, most of the vacuoles containing AP2XI-2-expressing parasites showed high DBL-positive staining. In contrast, when the parasites are maintained in an alkaline condition in the presence of IAA, only a few vacuoles exhibited faint DBL staining, and the presence of the bradyzoite-specific protein BAG1[23] was nearly undetectable (Fig. 2a, b). Intriguingly, over 95% of the AP2XI-2-depleted parasites appeared thinner or with a multinucleated phenotype. Using IMC1 as a marker of the mature and developing parasites, we observed that multinucleated parasites underwent endopolygeny division, resulting in a reduction in the IMC1 signal in the multinucleated maternal parasites; whereas the IMC1 fluorescence signal remained positive in the daughter parasites (Fig. 2a). Similar phenotypes were observed in the PruΔ*ap2XI-2* strain (Fig. 2a, b). In contrast, these morphological abnormalities were rarely observed when the parasites were maintained in a neutral medium with IAA (Fig. 1a).

### AP2XI-2 recapitulates the merozoite transcriptional program

To determine the transcriptomic changes attributable to AP2XI-2 deletion, the transcriptomes of AP2XI-2-mAID parasites cultured in

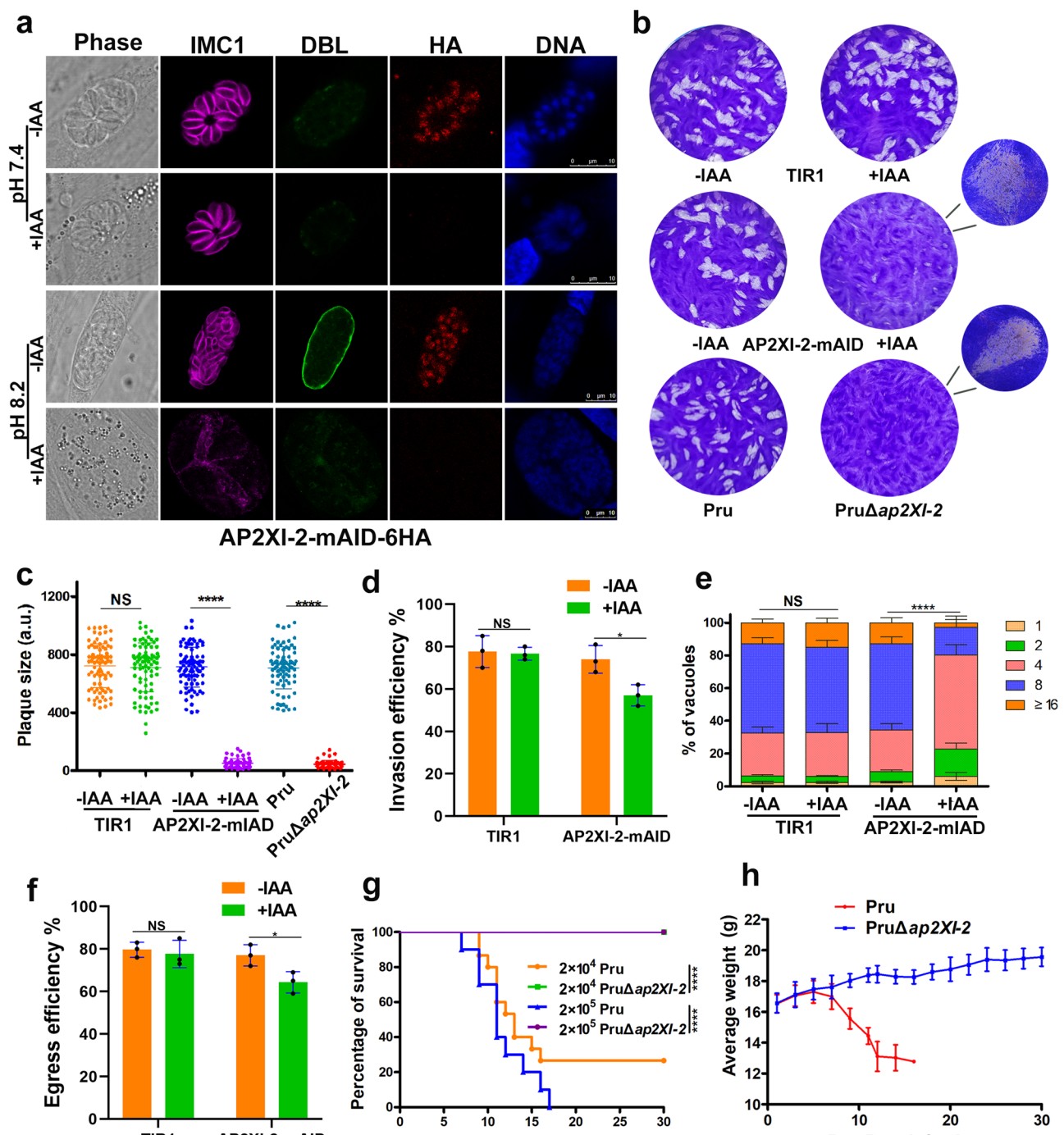

**Fig. 1 | AP2XI-2 is critical for the growth of *Toxoplasma gondii* in vitro and in vivo. a** AP2XI-2-mAID-6HA is localized to the nucleus in the tachyzoites (32 h post-infection) and bradyzoites (3 days post-infection) and is efficiently depleted when the parasites are treated with IAA (3-indoleacetic acid). Magenta, anti-IMC1; green, FITC-*Dolichos biflorus* lectin (FITC-DBL); red, anti-HA. Scale bar, 10 μm. **b** Representative images of the plaques formed by the indicated parasite strains grown in HFF monolayers for 9 days. **c** Relative size of the plaques detected in **b**. Data represents the mean ± SD from three independent experiments. Statistical significance was tested by two-tailed, unpaired *t* test, ****$p < 0.0001$. **d** Quantification of invasion of the indicated strains grown in HFF monolayers in the presence or absence of IAA. Data represents the mean ± SD from three independent experiments, analyzed by two-tailed, unpaired *t* test, *$p = 0.0234$. **e** Quantification

of the replication of the indicated strains grown in HFFs for 30 h in the presence or absence of IAA (added 2 h post-invasion). Data represents the mean ± SD from three independent experiments, analyzed by two-way ANOVA with Tukey multiple comparison test, ****$p < 0.0001$. **f** Quantification of the egress of the indicated strains grown in HFFs for 48-60 h in the presence or absence of IAA (added 2 h post-invasion). Data represents the mean ± SD from three independent experiments, analyzed by two-tailed, unpaired *t* test, *$p = 0.0365$. **g** Survival of C57BL/6 mice infected by $2 \times 10^4$ or $2 \times 10^5$ tachyzoites of the indicated strains. Survival curve of mice were compared using Gehan-Breslow-Wilcoxon test, $p < 0.0001$. ($n = 15$ mice/strain for the $2 \times 10^4$ dose; $n = 10$ mice/strain for the $2 \times 10^5$ dose). **h** The average body weight ± SD of the mice infected by $2 \times 10^5$ of the parental Pru or PruΔ*ap2XI-2* tachyzoites. ($n = 10$ mice / strain). Source data are provided as a Source data file.

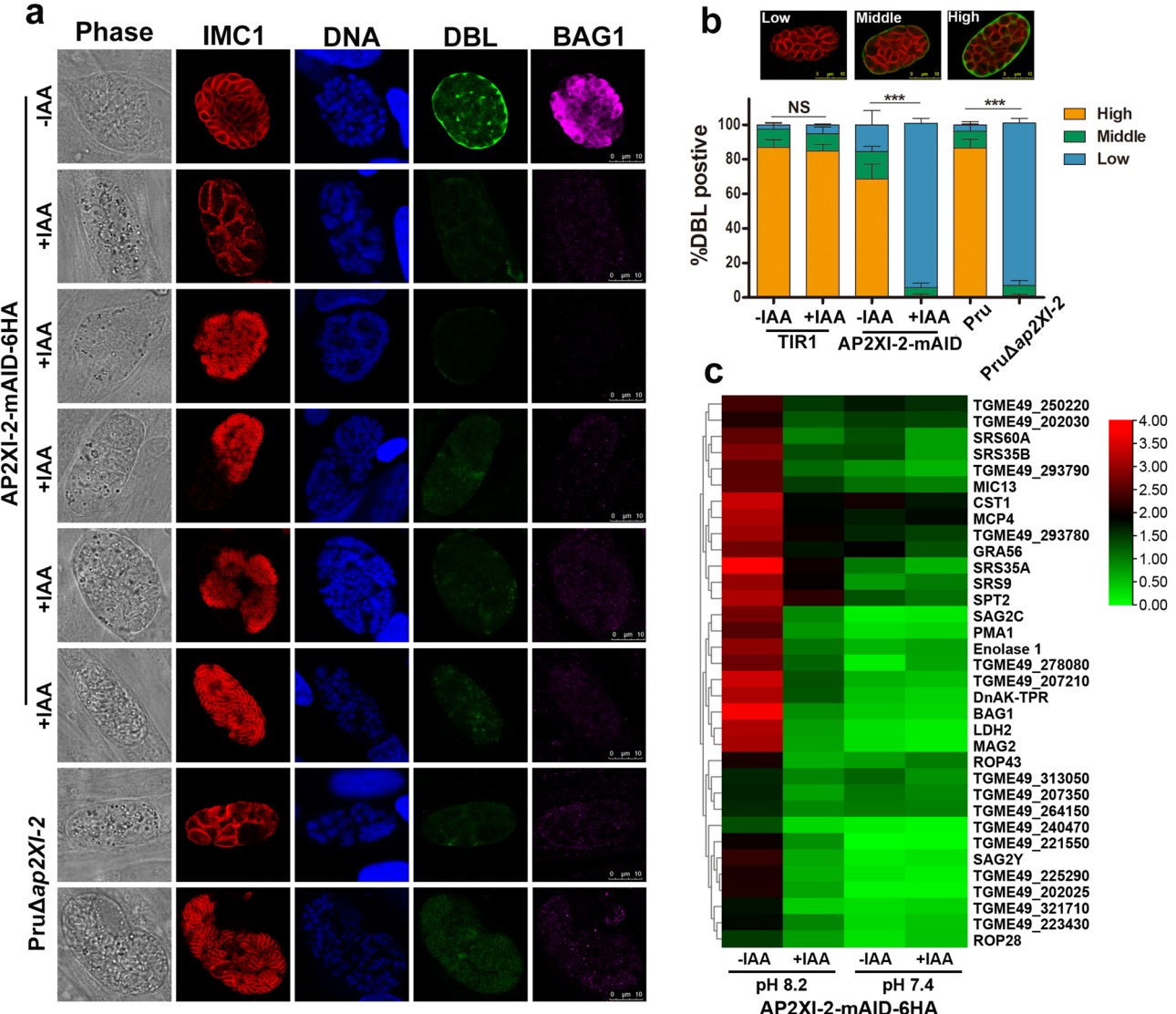

**Fig. 2 | Loss of AP2XI-2 blocks bradyzoite differentiation and results in severe morphological abnormalities under alkaline conditions. a** Representative images of the indicated parasite strains grown in an alkaline culture medium without $CO_2$ for 3 days for induction of bradyzoites. Red, anti-IMC1; green, FITC-DBL; magenta, anti-BAG1. Scale bar, 10 μm. **b** Quantification of bradyzoite differentiation in the indicated strains 3 days after exposure to alkaline stress. Data are presented as the mean ± SD from at least three independent biological replicates. The percentage of DBL-positive vacuoles was calculated based on at least 150 vacuoles per replicate. The percentage of vacuoles scoring "DBL-high" was used for comparison and analyzed by two-tailed, unpaired $t$ test, ***$p$ = 0.0002 (AP2XI-2-mAID -IAA vs +IAA), ***$p$ = 0.0001 (Pru vs PruΔ*ap2XI-2*). **c** Heat map showing the decreased expression of selected bradyzoite highly expressed genes after depletion of AP2XI-2 under alkaline conditions. The color scale indicates $\log_{10}$-transformed FPKM values. Source data are provided as a Source data file.

neutral and alkaline conditions with or without IAA were sequenced. Under bradyzoite-inducing conditions without IAA, as expected, the majority of bradyzoite markers, such as CST1, BAG1, LDH2, SRS9, and enolase 1, were significantly upregulated in AP2XI-2-expressing parasites. However, the expression of many bradyzoite-related genes was not significantly altered in the AP2XI-2-depleted strain grown under bradyzoite-inducing conditions with IAA treatment (Figs. 2c and 3a, and Supplementary data 1).

Interestingly, depletion of AP2XI-2 induced the expression of merozoite-specific genes such as GRA11A/B, MIC17A/B, and families A-E under both neutral and alkaline conditions[24,25] (Fig. 3a, b and Supplementary data 1). Comparative RNA-seq analysis using existing datasets generated from merozoites[25] or early and mature bradyzoites[26] revealed that 48.7% (152/312) of merozoite-specific genes were induced, while 33.3% (97/291) of the highly expressed bradyzoite genes were suppressed by depletion of AP2XI-2 under

alkaline conditions. Additionally, most of the upregulated merozoite stage-specific mRNA levels were further increased in the alkaline medium compared to those in the neutral medium (Supplementary Fig. 2a). On the other hand, many genes which are highly expressed in tachyzoites, such as SAG1, MIC2, ROP1 and GRA1, were further downregulated in the alkaline medium (Supplementary Fig. 2b). To verify that depletion of AP2XI-2 induced merozoite-primed pre-sexual commitment, we examined the expression of several merozoite reporter proteins (GRA11B, GRA81, SRS48Q, and MIC17A) by immunofluorescence microscopy. As expected, these merozoite proteins were detected after depletion of AP2XI-2 under both neutral and alkaline medium conditions (Supplementary Fig. 2c). However, the genes involved in the formation of gametes and oocysts such as PF16[14,27], HAP2[14,28], and SporoSAG[14,29] were not significantly upregulated (Fig. 3a, b and Supplementary data 1).

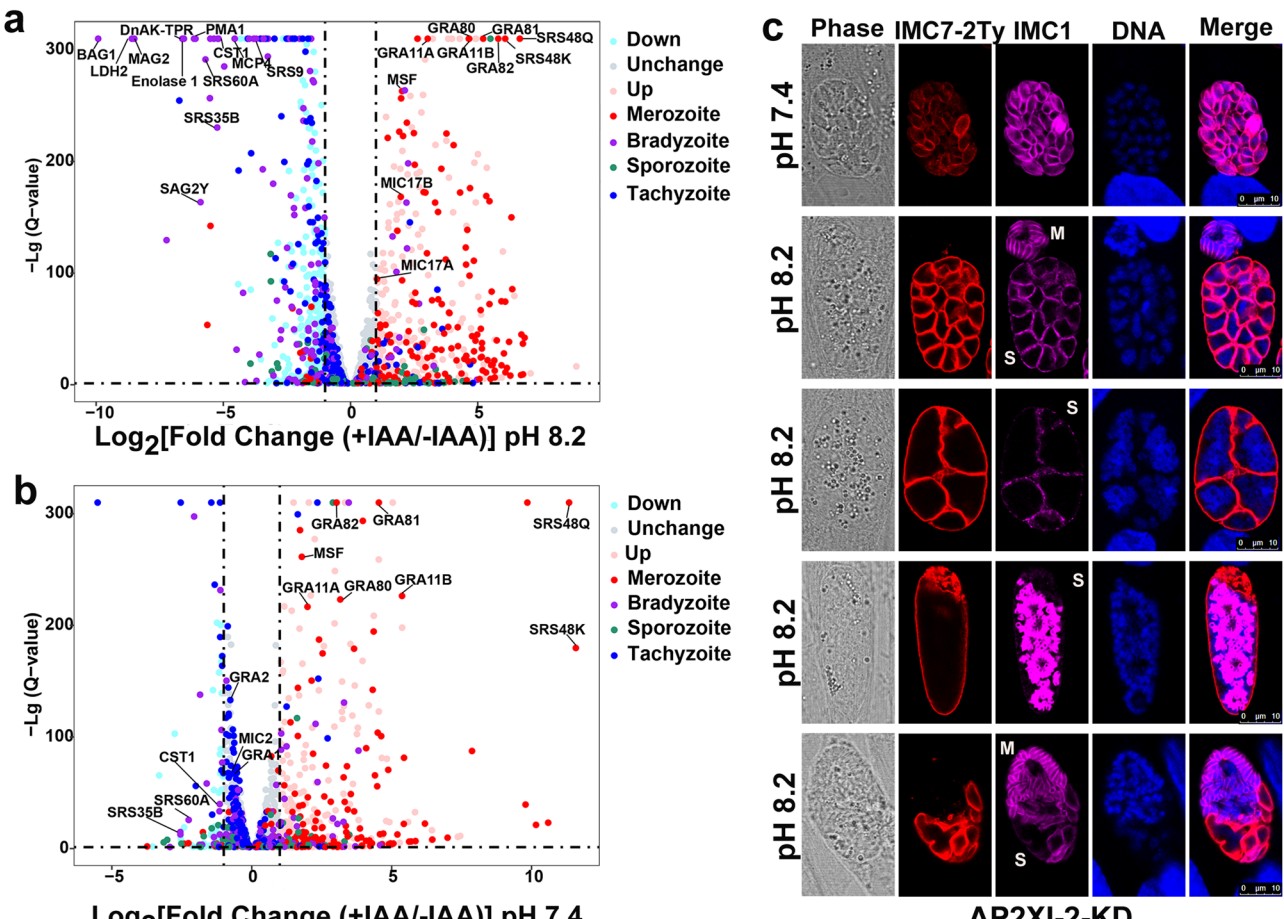

**Fig. 3 | Depletion of AP2XI-2 induces tachyzoites-to-merozoites switch under alkaline condition. a** Volcano plot showing the differentially expressed genes in the AP2XI-2-expressing and AP2XI-2-depleted strains grown under alkaline culture conditions for 3 days. The merozoite, bradyzoite, sporozoite and tachyzoite highly expressed genes were analyzed from Supplementary data 1. Data from four biological replicates were plotted and a fold change of ≥2.0 or ≤−2.0 with a p-value < 0.05 was considered statistically significant. **b** Volcano plot showing the differentially expressed genes in the AP2XI-2-expressing and AP2XI-2-depleted strains grown under neutral culture conditions. The merozoite, bradyzoite, sporozoite and tachyzoite highly expressed genes were analyzed from Supplementary data 1. Data from four biological replicates were plotted and a fold change of ≥2.0 or ≤−2.0 with a p-value < 0.05 was considered statistically significant. **c** Representative images of AP2XI-2 knockdown (KD) parasite strains grown in a neutral or an alkaline culture medium without $CO_2$ for 3 days for induction of merogony. IMC1 (magenta) specifically stains the (developing) merozoites, whereas IMC7-2Ty (red) is specific to the polyploid AP2XI-2-depleted schizonts. M merozoite, S schizont. Scale bar, 10 μm.

## Depletion of AP2XI-2 under alkaline conditions promotes switching from tachyzoite to merozoite features

Upon infection of cats, bradyzoites undergo merogony to produce schizonts and merozoites via endopolygenic reproduction[1]. Given the upregulated merozoite genes and downregulated tachyzoite genes, we speculated that the abnormal parasites may be schizonts and mature merozoites. To confirm this, we examined the morphological features of the AP2XI-2-depleted parasites growing in an alkaline medium. During merogony, the IMC1 and IMC7 proteins exhibit a dynamic expression pattern[30]. The parasites become multinucleated schizonts, resulting in a significantly reduced IMC1 fluorescence signal, while IMC7 remains at the periphery of the maternal parasite. After several cycles of endopolygeny, mature schizonts with fully developed merozoites are formed, which are strongly stained for IMC1 but are completely negative for IMC7[30]. Using these two IMC markers, we found that the AP2XI-2-depleted parasites grown in an alkaline medium undergo merogony with multinucleated schizonts and asynchronous division (Fig. 3c). Additionally, transmission electron microscopy (TEM) confirmed the multinucleated structures in the single maternal parasite, and the polyploid schizonts and mature merozoites were detected inside the same vacuole (Supplementary Fig. 2d). Immunofluorescence analysis further showed that proteins such as GRA1,

GRA2, MIC2, M2AP, ROP1, and SAG1, which are highly expressed in tachyzoites, were strongly suppressed in the polyploid schizonts and not detected in mature merozoites (Supplementary Fig. 2e). On the other hand, several house-keeping proteins, such as HSP60, β-tubulin, and ARO, the latter mediates the apical positioning of rhoptry organelles, can be detected in the mature merozoites[31,32] (Supplementary Fig. 2e), indicating that these house-keeping genes may play the same role in merozoites.

## AP2XI-2-depleted parasites undergoing merogony does not require bradyzoite stage

We examined whether the bradyzoite stage contributes to the merogony induced by the depletion of AP2XI-2 under alkaline conditions. Firstly, the AP2XI-2-expressing parasites were induced to activate the expression of bradyzoites-specific genes by culturing them in an alkaline medium for 2 days, followed by maintaining in neutral medium. After another 2 days of growth in the presence of IAA, the merogony phenotype was not observed (Supplementary Fig. 3a). Secondly, we examined the roles of AP2XI-2 in the PruΔ*bfd1* strain, which completely ablates bradyzoite formation[33]. As expected, no parasite vacuoles were markedly stained with DBL when Pru::AP2XI-2-mAID-6HAΔ*bfd1* parasites were maintained in an alkaline medium with

or without IAA for 3 days. However, polyploid schizonts and mature merozoites were also observed in the IAA-treated Pru::AP2XI-2-mAID-6HAΔ*bfd1* parasites, with a pattern similar to that observed in AP2XI-2-depleted parasites (Supplementary Fig. 3b). Moreover, the BFD1 was undetected in the AP2XI-2-depleted parasites under alkaline conditions, confirming that parasites deficient in AP2XI-2 did not form bradyzoites under alkaline conditions. The expression of AP2XI-2 was not affected by deletion of BFD1 (Supplementary Fig. 3c, d). Finally, we evaluated whether depletion of AP2XI-2 in the RHΔ*ku80*Δ*hxgprt*, which grows faster than the Pru strain and rarely differentiates into bradyzoites in vitro[34,35], induces the parasites to undergo merogony. Similar to Pru::AP2XI-2-mAID-6HA, the AP2XI-2 protein was rapidly degraded in the RH::AP2XI-2-mAID-6HA strain after the addition of IAA (Supplementary Fig. 3e, f). As expected, a few parasite vacuoles were faintly stained with DBL when RH::AP2XI-2-mAID-6HA was grown in an alkaline medium with or without IAA for 2 days, and depletion of AP2XI-2 in the RH strain did not result in merogony (Supplementary Fig. 3f). Taken together, these data indicate that merogony caused by depletion of AP2XI-2 under alkaline conditions does not mark the beginning of the bradyzoite stage, and that the slow growth of the parasites in the alkaline medium may contribute to the merogony process, as observed in the Pru strain and even in the PruΔ*bfd1* strain, which showed slower growth in alkaline medium (Supplementary Fig. 3g).

### Identification of proteins interacting with AP2XI-2

A previous study showed that MORC protein functions in a complex with AP2 transcription factors to recruit the histone deacetylase HDAC3 to prevent the aberrant expression of genes normally restricted to the parasite sexual stage[14]. To better understand the roles of AP2XI-2, we performed co-immunoprecipitation experiments using AP2XI-2-mAID-6HA bradyzoites obtained after alkaline induction for 3 days. Based on three independent experiments, in addition to AP2XI-2, we also detected the MORC and HDAC3 proteins, which have been reported previously[14] to be associated with AP2XI-2, and several other proteins (Supplementary data 2).

To investigate the potential role of the identified proteins in the parasite growth, we focused on the top enriched proteins that are also found in the MORC/HDAC3 complexes, as previously reported[14]. Among those proteins, AP2XII-1, TGME49_209500, TGME49_214140 and TGME49_275680 were further examined by using C-terminal mAID-6HA endogenous tagging in the PruΔ*ku80*Δ*hxgprt*::TIR1-3Flag strains. These four proteins were detected in the nucleus of tachyzoites and bradyzoites, and were efficiently degraded after addition of IAA (Fig. 4a and Supplementary Fig. 4a). Additionally, the number of DBL-positive vacuoles were significantly decreased when these strains were grown in an alkaline medium with IAA (Fig. 4b). Plaque assays revealed that depletion of AP2XII-1 and TGME49_275680 significantly affected the parasite growth, while depletion of TGME49_209500 or TGME49_214140 had a minor effect on the parasite growth (Fig. 4c, d). To further investigate whether these proteins are involved in the merozoite-primed pre-sexual commitment, we evaluated the expression of GRA81 in the AP2XII-1, TGME49_209500, TGME49_214140, and TGME49_275680-depleted strains. The results showed that GRA81 was expressed when AP2XII-1 was depleted in both neutral and alkaline medium (Supplementary Fig. 4b). Interestingly, some AP2XII-1-depleted parasites became larger while some became thinner and appeared to form merozoites (Fig. 4a and Supplementary Fig. 4b). In contrast, TGME49_209500, TGME49_214140, and TGME49_275680-depleted parasites appeared normal (Supplementary Fig. 4a) and the GRA81 was not expressed in the TGME49_214140-depleted parasites and occasionally expressed in the TGME49_209500 or TGME49_275680-depleted parasites under alkaline culture conditions (Supplementary Fig. 4b).

Transcriptomes of AP2XII-1-mAID strain under neutral and alkaline medium conditions, with or without IAA, showed that depletion of AP2XII-1 induced the activation of merozoite-specific genes, which was confirmed by IFA (Supplementary Fig. 4b-e), and resulted in the suppression of typical tachyzoite highly expressed genes (Supplementary Fig. 4f). Similar to the transcriptomic changes detected in AP2XI-2-mAID, the merozoite-specific genes of AP2XII-1-depleted parasites were expressed at a much higher level in the alkaline medium than that in the neutral medium (Supplementary Fig. 4e). Interestingly, the expression of AP2XII-1, TGME49_209500, and TGME49_275680 was not detected in mature merozoites induced by the depletion of the AP2XI-2, and the expression of AP2XI-2 was not detected in the AP2XII-1-depleted mature merozoites (Fig. 4e), while TGME49_214140 was expressed in mature merozoites. Using CUT&Tag method, we found that AP2XI-2 and AP2XII-1 are co-located near the transcriptional start sites of merozoite-specific genes which points out to enrichment for MORC/HDAC3 complexes (Fig. 4f, g, Supplementary Fig. 4g, and Supplementary data 3). MORC occupancy at these merozoite-specific genes decreased upon depletion of AP2XI-2. These findings suggest that AP2XI-2 and AP2XII-1 are responsible for recruiting the MORC/HDAC3 complex to repress merozoite-specific genes.

### AP2XI-2 and AP2XII-1 play different roles in the pre-sexual stages

Given that depletion of AP2XI-2 and AP2XII-1 can induce parasites to undergo merogony in an alkaline medium, we investigated whether these two proteins play similar roles during merogony. We cultured AP2XI-2-mAID and AP2XII-1-mAID strains under alkaline culture conditions for 4 days and used different protein markers to investigate the merogony process. Over this period, AP2XI-2-depleted parasites exhibited a more pronounced merogony process, characterized by larger and more multinucleated maternal parasites, in comparison to AP2XII-1-depleted parasites (Fig. 5 and Supplementary Fig. 5). After one day of incubation in an alkaline culture medium, over 95% of AP2XI-2-depleted parasites increased in size with divided nuclei, as confirmed by staining using centrosome, kinetochore, centromere and apicoplast markers, but they did not form daughter zoites (Fig. 5a, Supplementary Fig. 5a, and Supplementary Fig. 6). In contrast, over 95% of AP2XII-1-depleted parasites underwent the first round of merogony, resulting in the formation of thinner zoites, that were negative for IMC7 staining (Fig. 5b and Supplementary Fig. 5b). On day 2 of exposure to the alkaline condition, the AP2XII-1-depleted thinner zoites continued to undergo merogony, and a single AP2XII-1-depleted maternal parasite often produced 8 to 16 zoites, whereas most AP2XI-2-depleted parasites produced large schizonts, many of which contained ≥ 32 nuclei (Fig. 5 and Supplementary Fig. 5). After 3 to 4 days, some AP2XI-2-depleted parasites became mature merozoites while others continued to undergo merogony. On the other hand, AP2XII-1-depleted zoites continued the merogony process, eventually producing mature merozoites (Fig. 5 and Supplementary Fig. 5). The mature merozoites lost the ability to reinvade HFF cells even under neutral medium conditions. Interestingly, AP2XII-1-depleted parasites cultured in neutral medium also underwent merogony, typically resulting in 4 to 8 zoites in a single maternal parasite, which was fewer than the zoites produced under alkaline culture conditions (Supplementary Fig. 7a, b).

Interestingly, we found that the ratio of AP2XII-1-depleted parasites stained for DBL and BAG1 was significantly higher than that observed in the AP2XI-2-depleted parasites (Supplementary Fig. 7c), and RNA-Seq confirmed that the abundance of typical bradyzoite highly expressed transcripts were significantly higher in the AP2XII-1-depleted parasites than in the AP2XI-2-depleted parasites under alkaline culture conditions (Supplementary Fig. 7d). These results suggest that the extent to which depletion of AP2XI-2 caused the parasites to undergo merogony was more pronounced than the effect of AP2XII-1 depletion under alkaline culture conditions.

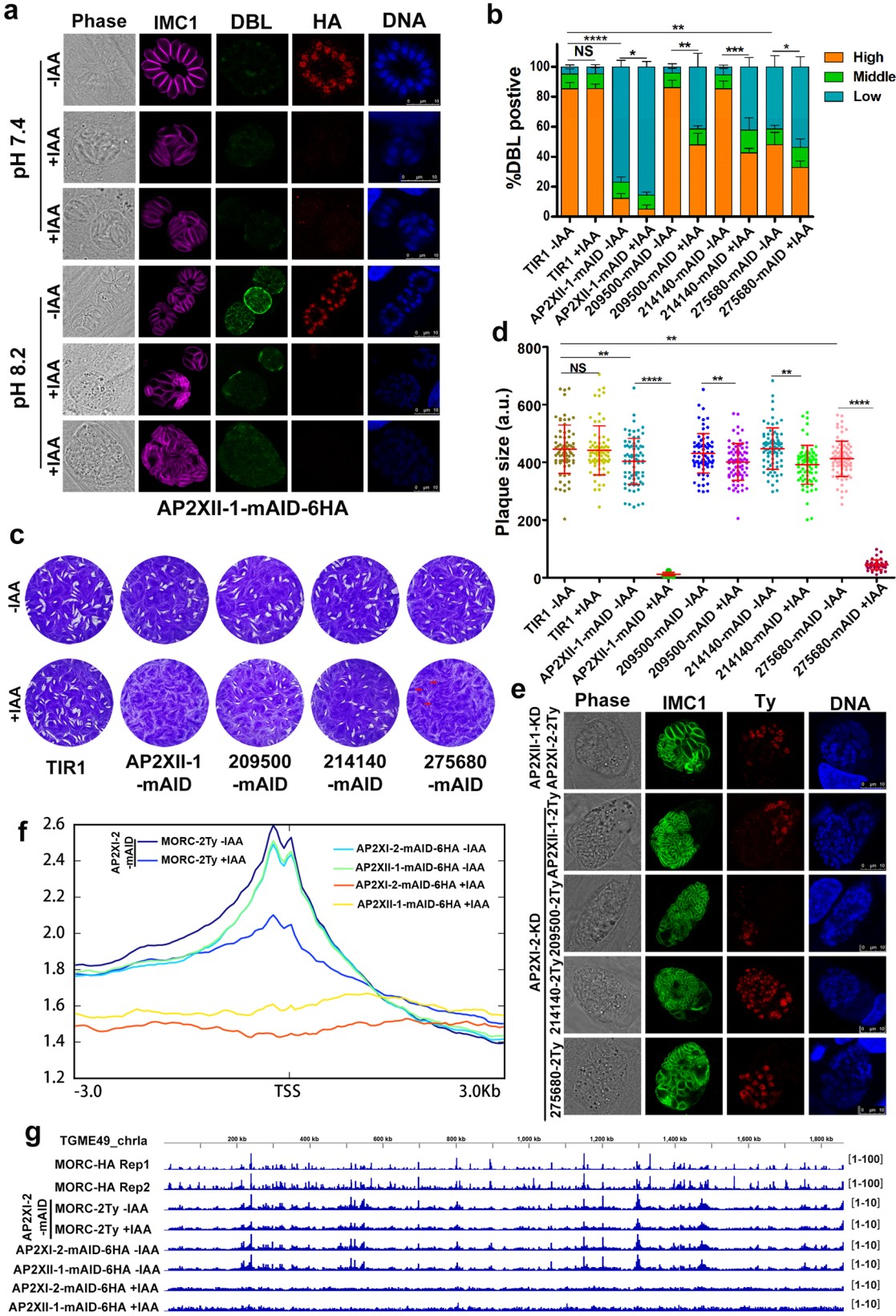

AP2XII-1-mAID-6HA

Surprisingly, co-depletion of AP2XI-2 and AP2XII-1 under neutral or alkaline medium conditions increased the merogony, often resulting in 8 to 16 zoites in a single maternal parasite, compared to depletion of only AP2XII-1 (Fig. 6). However, the effect was less pronounced than that observed in the AP2XI-2-depleted parasites grown in an alkaline medium, even when the AP2XI-2 and AP2XII-1 co-depleted

parasites were cultured in an alkaline medium (Fig. 6). These results suggest that AP2XI-2 may primarily control the merogony process, given its notable role in merogony, while AP2XII-1 may contribute to merozoite maturation.

The disruption of TGME49_209500 in the AP2XI-2-depleted parasites also induced merogony and production of mature

**Fig. 4 | AP2XII-1, an interacting partner of the AP2XI-2 complex, also silences merozoite-primed pre-sexual commitment. a** AP2XII-1-mAID-6HA localizes to the nucleus in tachyzoites (32 h post-infection) and bradyzoites (3 days post-infection) and is efficiently depleted when the parasites are treated with IAA. Magenta, anti-IMC1; green, FITC-DBL; red, anti-HA. Scale bar, 10 μm. **b** Quantification of bradyzoite differentiation in the indicated strains following 3 days of exposure to an alkaline culture medium. Data are presented as the mean ± SD from at least three independent biological replicates and the percentage of DBL-positive vacuoles was calculated based on a minimum of 150 vacuoles per replicate. The percentage of vacuoles scoring "DBL-high" was used for comparison and analyzed by two-tailed, unpaired $t$ test, *$p = 0.0379$ (AP2XII-1-mAID, -IAA vs +IAA), **$p = 0.0020$ (209500-mAID, -IAA vs +IAA), ***$p = 0.0003$ (214140-mAID, -IAA vs +IAA), *$p = 0.0449$ (275680-mAID, -IAA vs +IAA), **$p = 0.0022$ (TIR1 -IAA vs 275680-mAID -IAA), ****$p < 0.0001$ (TIR1 -IAA vs AP2XII-1-mAID -IAA). **c** Representative images of the plaques formed by the indicated parasites grown in HFF monolayers for 8 days.

**d** Relative size of the plaques detected in **c**. Data represents the mean ± SD from three independent experiments. Statistical significance was tested by two-tailed, unpaired $t$ test, **$p = 0.0019$ (TIR1 -IAA vs AP2XII-1-mAID -IAA), **$p = 0.0074$ (209500-mAID -IAA vs +IAA), **$p = 0.0014$ (214140-mAID -IAA vs +IAA), **$p = 0.0067$ (TIR1 -IAA vs 275680-mAID -IAA), ****$p < 0.0001$. **e** The subcellular location of the indicated proteins in the mature merozoites. The indicated parasites were allowed to infect HFFs for 4 h followed by incubation in an alkaline medium without $CO_2$ for 3 days with IAA. Green, anti-IMC1; red, anti-Ty; Scale bar, 10 μm. **f** Genome-wide analysis reveals that AP2XI-2, AP2XII-1 and MORC associate with the transcriptional start sites (TSS) of the genes. The indicated parasites treated with or without IAA for 48 h were used to assess the genomic loci of AP2XI-2, AP2XII-1 or MORC. **g** Integrated genome browser view of AP2XI-2, AP2XII-1 and MORC enrichment across *T. gondii* chromosome Ia. The data of MORC-HA Rep1 and MORC-HA Rep2 were obtained from the GEO under accession number GSE136060[14]. Source data are provided as a Source data file.

merozoites under neutral or alkaline conditions. The pattern of merogony resembled that observed in AP2XI-2 and AP2XII-1 co-depleted parasites (Fig. 6c and Supplementary Fig 8). Disruption of TGME49_275680 in the AP2XI-2-depleted parasites also induced merogony in both neutral and alkaline conditions but the merogony was less pronounced than that observed in the AP2XI-2-depleted parasites grown in an alkaline medium (Fig. 6c and Supplementary Fig 8). However, disruption of TGME49_214140 in AP2XI-2-depleted parasites did not cause merogony under neutral culture conditions but decreased the extent of merogony and merozoites formation in an alkaline culture (Fig. 6c and Supplementary Fig 8). These data indicate that the merozoite-primed pre-sexual commitment is a complex process involving several regulators.

## Depletion of AP2XI-2 or AP2XII-1 activates several AP2 factors

The expression of genes typically highly expressed in bradyzoites and tachyzoites were decreased in the absence of AP2XI-2 or AP2XII-1 when parasites were exposed to alkaline stress. Our data also showed that the transcriptional level of several AP2 transcriptional factors increased when AP2XI-2 or AP2XII-1 was depleted (Supplementary data 4). Therefore, we hypothesized that there may be secondary transcriptional regulators guiding these transcriptional changes. To confirm this, we examined the expression patterns of these AP2 proteins in the AP2XI-2-depleted parasites by IFA. As expected, at least nine AP2 factors were expressed when the AP2XI-2-depleted parasites were grown in an alkaline medium (Fig. 7a). It is worth noting that AP2IV-3 which acts as an activator[12] and AP2IX-9 that serves as a repressor[17] were also expressed in the bradyzoite stage, hinting to their involvement in the regulation of cyst formation (Supplementary Fig. 9a, b). Interestingly, these nine AP2 factors exhibited different expression patterns during merogony. The AP2IV-3, AP2IV-2 and AP2VIIa-1 proteins were detected in mature merozoites. AP2X-3 appeared during merogony and increased in mature merozoites. In contrast, AP2IX-1 was detected during merogony, but has decreased and became undetectable in mature merozoites. TGME49_215895, an AP2 domain-containing protein, was slightly detected in the polyploid schizonts and mature merozoites. AP2X-2 was detected in the larger polyploid schizonts, whereas AP2Ib-1 and AP2IX-9 were occasionally detected in the AP2XI-2-depleted parasites, but not in a parasite stage-specific manner (Fig. 7a). Similar results were observed in the AP2XII-1-depleted parasites, except that TGME49_215895 was undetected, and AP2X-2 was occasionally detected in some small polyploid schizonts (Supplementary Fig. 9c).

To further investigate the role of these AP2 factors during merogony, we generated AP2 knockouts in the Pru::AP2XI-2-mAID-6HA strains. Surprisingly, we were unable to disrupt the AP2X-3 in the Pru::AP2XI-2-mAID-6HA strain. Therefore, we added the AID-Ty fusion to the C-terminus of the AP2X-3 in the Pru::AP2XI-2-mAID-6HA or PruΔ*ku80*Δ*hxgprt*::TIR1-3Flag strain. The disruption of eight AP2

factors did not significantly affect the ability of the AP2XI-2-depleted parasites to undergo merogony and produce mature merozoites under alkaline conditions, although disruption of AP2IX-1 and AP2X-2 slightly affected the ratio of the mature merozoites (Fig. 7b, c). As expected, due to the important role of AP2X-3 in the parasite growth, depletion of AP2X-3 in the AP2XI-2-depleted Pru parasites resulted in the production of abnormal schizonts and merozoites; however, AP2X-3 protein was not detected in the tachyzoite or bradyzoite stage (Fig. 7d, e). The disruption of these AP2s in the AP2XII-1-depleted strains also did not affect merogony (Supplementary Fig. 9d). These results suggest that these AP2s are less likely to be indispensable for *T. gondii* to undergo merogony and produce mature merozoites under the study conditions.

## Discussion

Sexual development is necessary for the completion of the life cycle of *T. gondii*; however, the repertoire of the transcriptional regulators underlying sexual commitment in this parasite remains understudied. Transcription factors such as the AP2 are involved in the regulation of key processes during the parasite development and stage transformation[12,15–19]. Here, we identify two AP2 transcription factors, AP2XI-2 and AP2XII-1, as indispensable repressors of the gene pathway controlling *T. gondii* pre-sexual development. We show that AP2XI-2 and AP2XII-1 are expressed in both tachyzoite and bradyzoite stages, and positively regulate the asexual phase of reproduction and tachyzoite-to-bradyzoite differentiation through inhibition of merozoite-specific genes. Conversely, during sexual reproduction, AP2XI-2 and AP2XII-1 are downregulated to promote the expression of genes involved in merogony.

The epigenetic repressor MORC partners with HDAC3 and at least 12 AP2 transcription factors to maintain the bradyzoite stage and sexual stage genes in a continuously repressed chromatin state[14]. Depletion of MORC or inhibition of HDAC3 induces the expression of the bradyzoite, pre-sexual, and sexual stage-specific genes. Among the AP2 factors, AP2XI-2 and AP2XII-1 bind with the MORC/HDAC3 complexes in the tachyzoite stage[14]. In our study, AP2XI-2 and AP2XII-1 were continuously expressed in the tachyzoites and bradyzoites, and were also associated with MORC/HDAC3 complexes during the bradyzoite stage. Depletion of AP2XI-2 or AP2XII-1 induced the expression of genes specific to the pre-sexual stage. Conversely, a larger number of genes highly expressed in tachyzoite stage were silenced. Interestingly, when the AP2XI-2 or AP2XII-1-depleted parasites were cultured in an alkaline medium, the transcriptional level of the genes specific to pre-sexual stage were further increased and the abundance of tachyzoite highly expressed transcripts were further decreased.

Regarding the parasite morphology, the AP2XI-2-depleted parasites resembled the tachyzoites, whereas the AP2XII-1-depleted parasites appeared thinner and more likely to be merozoites under neutral conditions[36,37]. When parasites were maintained in alkaline conditions,

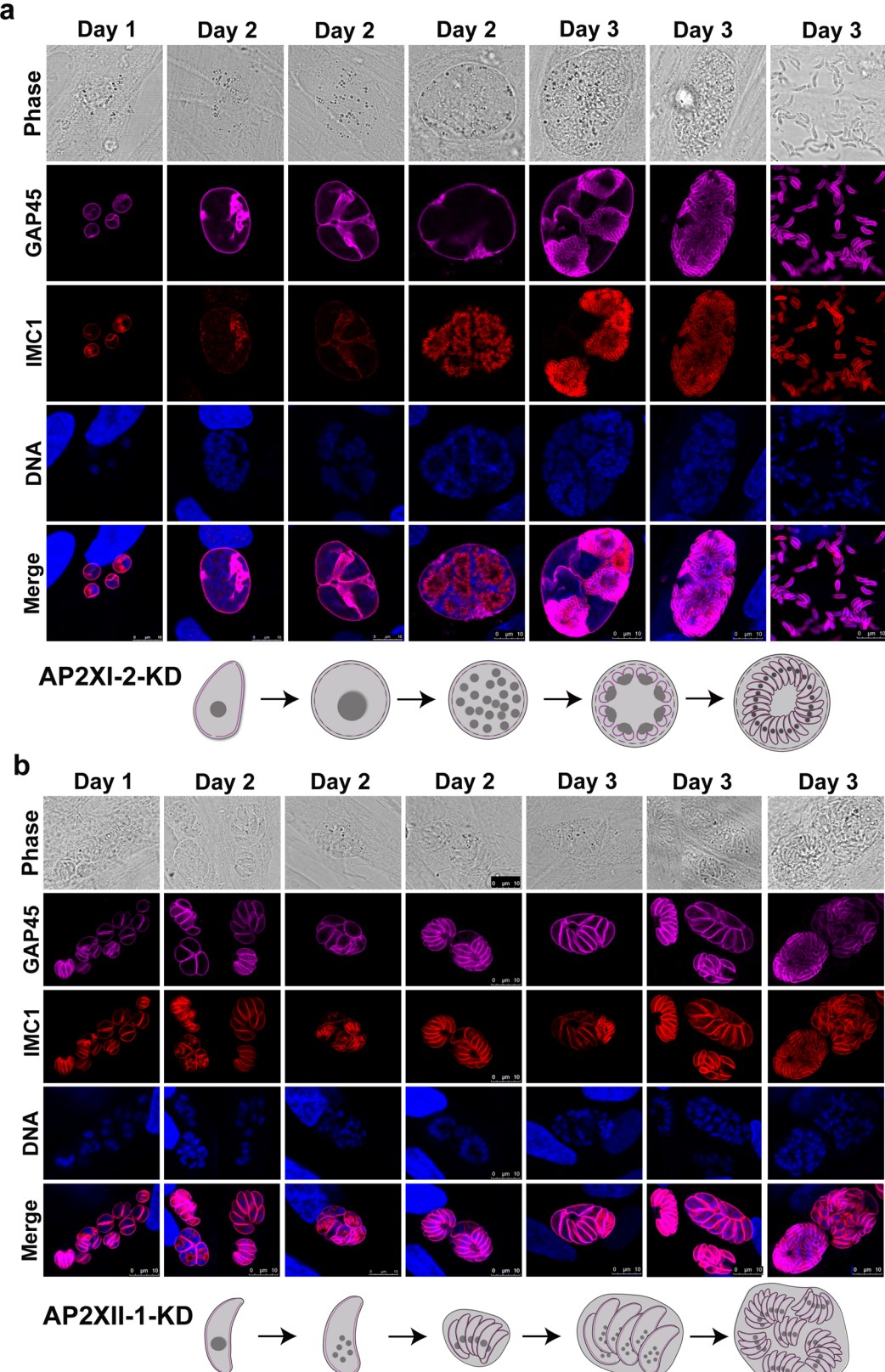

**Fig. 5 | Characterization of merozoite formation after depletion of AP2XI-2 or AP2XII-1 under alkaline conditions.** Representative images of AP2XI-2 (**a**) or AP2XII-1 (**b**) knock-down (KD) parasites grown in an alkaline medium without $CO_2$ over 3 days for induction of merozoites. Magenta, anti-GAP45; red, anti-IMC1. Scale bar, 10 μm.

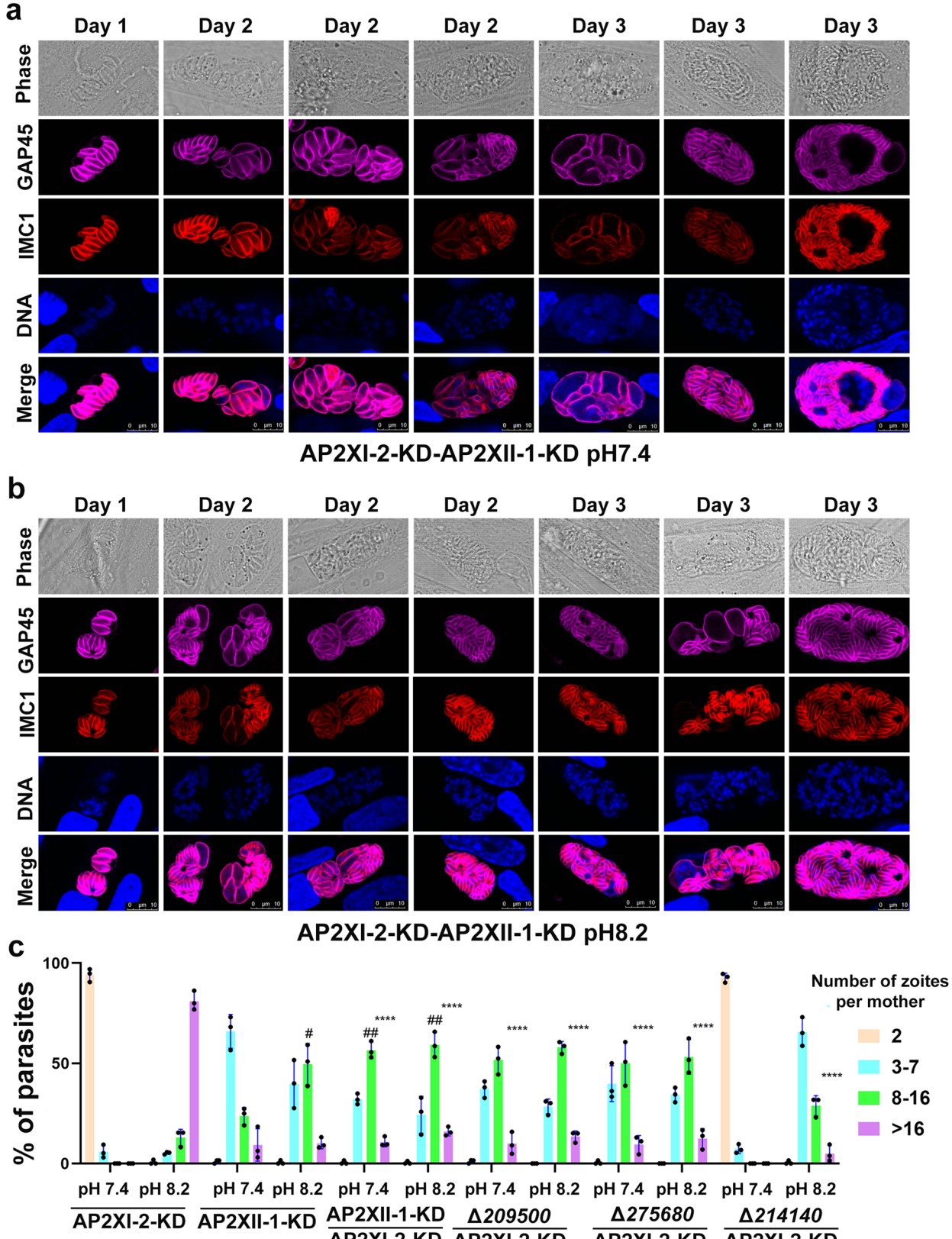

the AP2XI-2 or AP2XII-1-depleted parasites were unable to induce bradyzoite genes and failed to form cysts. Interestingly, most of the AP2XI-2-depleted parasites became multinucleated and divided by endopolygeny forming more than 32 daughters in one AP2XI-2-depleted maternal parasite and the newly formed parasites were thinner, whereas the AP2XII-1-depleted parasites produced fewer than 16 daughter parasites. Using the IMC1 and IMC7 markers[30], we found that AP2XI-2-depleted parasites undergo merogony and produce mature merozoites under alkaline conditions but not in a neutral medium. In comparison, the commitment to merogony in AP2XII-1-depleted parasite was detected in both neutral and alkaline conditions, with a greater extent in the alkaline medium. Interestingly, the extent

**Fig. 6 | Characterization of the roles of AP2XI-2 interacting proteins in the AP2XI-2-depleted parasites. a, b** Representative images of AP2XI-2 and AP2XII-1 co-depleted parasites grown in neutral (**a**) or alkaline medium (**b**) over 3 days for induction of merozoites. Magenta, anti-GAP45; red, anti-IMC1. Scale bar, 10 μm. **c** Quantitative analysis of the budding patterns in the indicated strains treated with IAA for 3 days. Data are presented as the mean ± SD from three independent biological replicates. The percentage of zoites per maternal parasite was calculated based on at least 50 parasites per replicate. Statistical significance was tested by two-tailed, unpaired *t* test. *Compared with the percentage of >16 zoites in the AP2XI-2-KD cultured under an alkaline medium, ****$p < 0.0001$; #compared with the percentage of 8–16 zoites in the AP2XII-1-KD cultured under a neutral medium, #$p = 0.016$ for AP2XII-1-KD cultured under an alkaline medium; ##$p = 0.0024$ for AP2XI-2-KD-AP2XII-1-KD cultured under a neutral medium; ##$p = 0.0013$ for AP2XI-2-KD-AP2XII-1-KD cultured under an alkaline medium.

of merogony in the AP2XI-2 and AP2XII-1 co-depleted parasites was greater than that observed in the AP2XII-1-depleted parasites, but was less than that detected in the AP2XI-2-depleted parasites under alkaline conditions, suggesting that AP2XI-2 may predominantly represses the genes involved in merogony, whereas AP2XII-1 may be involved in silencing genes involved in merozoite maturation. In addition to AP2XI-2 and AP2XII-1, two other nuclear factors TGME49_209500 and TGME49_275680 also associated with the MORC/HDAC3 complexes[14], were involved in the regulation of merogony. However, depletion of one of these two factors alone failed to induce merogony. These data indicate that a complex network of transcriptional repressors is involved in regulating merozoite-primed pre-sexual commitment in *T. gondii*.

The AP2XI-2-depleted parasites underwent endopolygeny to produce mature merozoites under alkaline conditions. During the merogony process, proteins such as SAG1, MIC2, and ROP1, which are highly expressed in tachyzoites and play important roles in the invasion or gliding motility of tachyzoites, were markedly decreased and not detected in mature merozoites. This reduction may account for the significant reduction in the plaque formation after the depletion of AP2XI-2 or AP2XII-1. In contrast, merozoite-specific transcripts, such as MIC17A/B and GRA11A/B, were significantly expressed, although only about half of the known merozoite mRNAs were induced, including the weak expression of the dominant merozoite-specific AP2, after the depletion of AP2XI-2 under alkaline conditions. This could be attributed to the fact that only 10-15% of the AP2XI-2-depleted parasites produced mature merozoites when the samples were collected for RNA-Seq, while the remaining AP2XI-2-depleted parasites continued to undergo merogony. These results suggest that the depletion of AP2XI-2 or AP2XII-1 results in transcriptional reprogramming, switching tachyzoite to merozoite, which is compatible with early stages of *T. gondii* sexual development that exclusively occurs in the cat intestinal epithelial cells.

Kittens typically excrete oocysts 3-10 days after ingesting bradyzoites or tissue cysts. However, the prepatent period can be longer (≥ 18 days) following ingestion of tachyzoites[38–41]. The additional time from days 11 to 18 post-ingestion of tachyzoites may be required for the switch of tachyzoites to bradyzoites[42,43]. Therefore, we wondered whether AP2XI-2-depleted tachyzoites need to differentiate into bradyzoites before undergoing merogony to produce merozoites. Surprisingly, deletion of the bradyzoite master gene *bfd1* in the AP2XI-2-depleted Pru strain did not affect the parasite's ability to undergo merogony, suggesting that the bradyzoite stage was not necessary for the AP2XI-2-depleted parasites to undergo merogony under alkaline conditions. Transcriptome analysis showed significant reduction in the abundance of highly expressed tachyzoite transcripts, but the abundance of bradyzoite-associated transcripts did not significantly increase when the AP2XI-2-depleted parasites were cultured under alkaline conditions. Intriguingly, the expression of tachyzoite highly expressed genes was also slightly decreased in the *bfd1* knockout strain, and the growth of *bfd1* knockout was relatively slow under alkaline medium compared to that in the neutral medium[33]. On the other hand, depletion of AP2XI-2 in the type I RH strain failed to induce merogony under alkaline conditions. It is worth noting that the parental RH and Pru strains used in this study are not developmentally competent and have undergone many passages in vitro, which can affect the parasite ability to complete the life cycle in cats. The reason

why AP2XI-2 depletion in type II Pru strain but not in type I RH strain enabled the parasite to undergo merogony and produce merozoites under the conditions tested in the present study warrants further investigation. Additionally, the roles of AP2XI-2 in other type II strains, such as the developmentally competent ME49 strain, or in different genotypes, also warrant further investigation.

Reassuringly, while this manuscript was under revision, Antunes et al.[44] showed that the AP2XI-2 and AP2XII-1 co-depleted RH parasites can undergo several rounds of merogony to produce merozoites. This may be attributed to the slow growth of the AP2XI-2 and AP2XII-1 co-depleted RH parasites, where the transcriptomic data showed significant reduction in the expression of tachyzoite highly expressed genes compared with those detected in parasites individually deficient in AP2XI-2 or AP2XII-1[44]. Additionally, another recent study has identified AP2XII-1 as a merogony repressor[45]. In *Plasmodium*, AP2-G works with AP2-G2 to control the development of the sexual stage[46,47]. AP2-G acts as a master regulator to activate gametocyte-specific genes, while AP2-G2 represses the development of asexual-stage genes[48]. These findings suggest that the alkaline medium may silence the expression of tachyzoite-specific genes to favor commitment to merogony, and that secondary transcriptional factors may exist during merogony to suppress the expression of tachyzoite-specific genes.

Indeed, the depletion of AP2XI-2 or AP2XII-1 induced the expression of several secondary AP2 factors. Among those AP2 factors, AP2IX-1, mainly expressed in the polyploid schizonts, is known to suppress the tachyzoite-specific SAG1 expression when transiently expressed in tachyzoites[49]. Surprisingly, disruption of AP2IX-1 in the AP2XI-2-depleted parasite slightly affected the number of mature merozoites suggesting that other transcription factors may act synergistically with AP2IX-1 to suppress the expression of tachyzoite-specific genes during merogony. The bradyzoite transcriptional repressor AP2IX-9, which significantly reduces tissue cyst formation when overexpressed, was also upregulated in the AP2XI-2- or AP2XII-1-depleted parasites, suggesting that degradation of AP2XI-2 or AP2XII-1 may induce AP2IX-9 to prevent the merozoites from converting back into bradyzoites[17], although the disruption of AP2IX-9 in the AP2XI-2- or AP2XII-1-depleted parasites did not affect merozoite formation. Disruption of other AP2 factors, including AP2IV-3, AP2IV-2, AP2VIIa-1, and AP2X-3 whose expression was mainly detected in mature merozoites as well as TGME49_215895, AP2X-2, and AP2Ib-1, whose expression was slightly increased, did not affect the pre-sexual commitment in the AP2XI-2-depleted parasite, indicating that these factors may be involved in other processes, such as gamete development or oocyst formation. These results suggest that *T. gondii*'s merozoite-primed pre-sexual commitment is regulated by intricate interactions between activators and suppressors, which warrants further study.

In summary, our results show that depletion of AP2XI-2 or AP2XII-1 in *T. gondii* type II Pru strain severely affects the lytic cycle of the tachyzoites and induces *T. gondii* to undergo merogony and produce mature merozoites. Although these in vitro produced merozoites are incapable of re-infecting HFF cells and did not form microgametocytes and macrogametocytes, further characterization of their function would benefit from the use of feline intestinal epithelial cells and/or a nonfeline intestinal cell line deficient in delta-6-desaturase that confers a linoleic acid-rich environment and supports *T. gondii* sexual development[50]. The identification of several

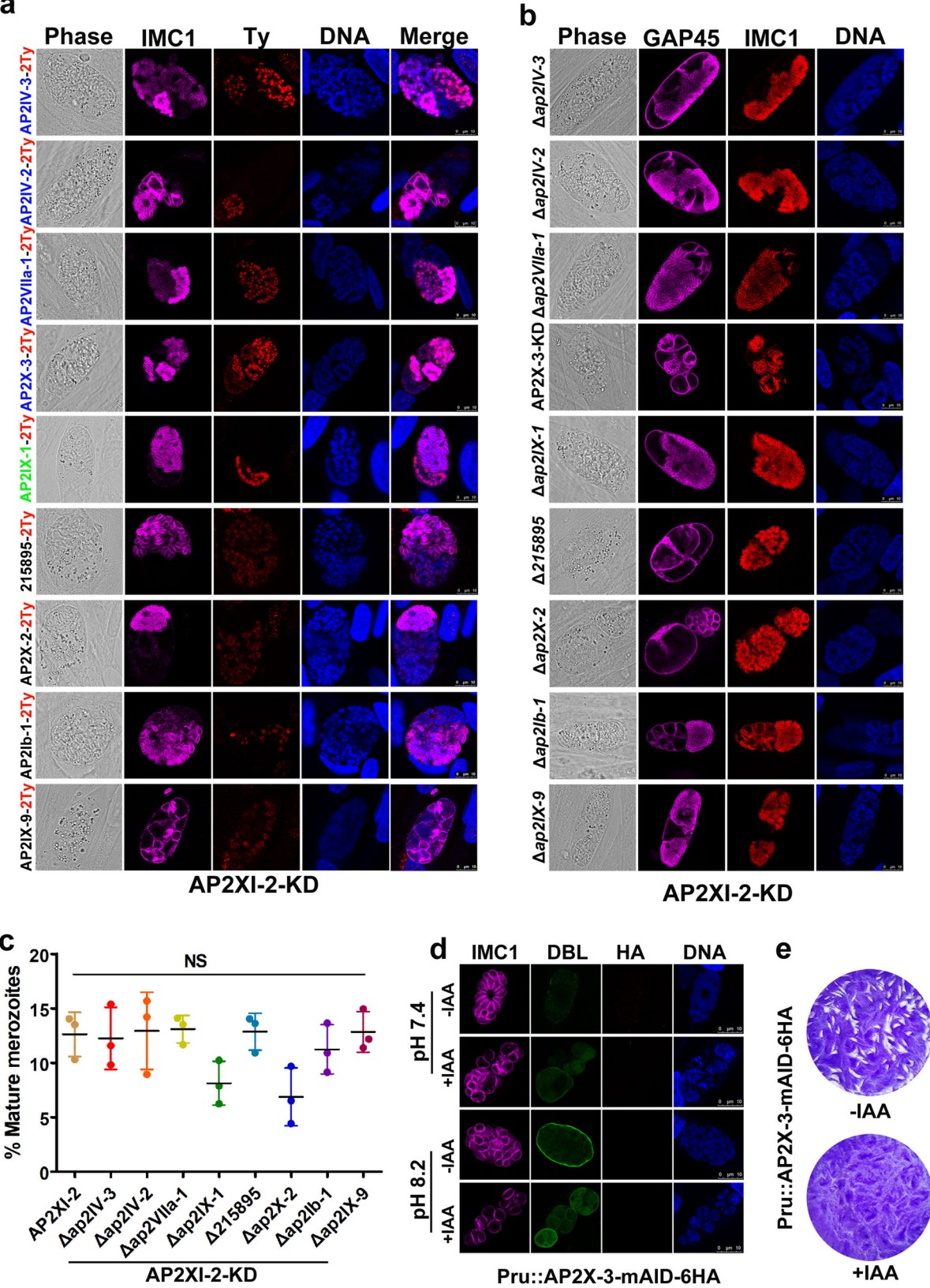

interacting or secondary transcriptional regulators that guide merogony sheds new light on the intricate regulatory processes controlling sexual commitment in *T. gondii*. By establishing AP2XI-2 and/or AP2XII-1-depleted Pru parasites, our study offers alternative in vitro models to study *T. gondii* merogony without the need to infect kittens. The data significantly advances our understanding of merogony biology and provides new insight into possible strategies for blocking *T. gondii* sexual development and the transmission of infection from the definitive feline host to the intermediate hosts.

**Fig. 7 | Several secondary transcriptional regulators are activated to regulate developmental stage transition after depletion of AP2XI-2. a** Confocal images showing the expression of secondary transcriptional regulators during merogony after depletion of AP2XI-2 under alkaline conditions for 3 days. The secondary transcriptional AP2 factors shown in blue and green colors on the left side of the figure denote those mainly expressed in mature merozoites and schizonts, respectively. Factors shown in black color denote those with slight increase upon depletion of AP2XI-2. Magenta, anti-IMC1; red, anti-Ty. Scale bar, 10 μm. **b** Representative images of the indicated parasite strains grown in an alkaline medium for 3 days for induction of merozoites. Magenta, anti-GAP45; red, anti-IMC1. Scale bar, 10 μm. **c** Quantification of mature merozoites in the indicated strains following 3 days of exposure to alkaline stress. Data represents the mean ± SD from three independent experiments, analyzed by two-tailed, unpaired $t$ test, NS, no significant. **d** AP2X-3-mAID-6HA was undetected in tachyzoites (32 h post-infection) and bradyzoites (3 days post-infection). Magenta, anti-IMC1; green, FITC-DBL; red, anti-HA. Scale bar, 10 μm. **e** Representative images of the plaques formed by AP2X-3-mAID-6HA parasites grown in HFF monolayers for 8 days with or without IAA.

## Methods

### Parasite strains, culture conditions, and transfection
Parasite strains used in this study, including RHΔ*ku80Δhxgprt*::TIR1-3Flag, PruΔ*ku80Δhxgprt*::TIR1-3Flag, and PruΔ*ku80Δhxgprt*, were maintained at 37 °C and 5% $CO_2$ in confluent monolayers of human foreskin fibroblast (HFF) cells, as previously described[51]. Tachyzoites were mechanically released from heavily infected HFFs by passage through a 27-gauge needle, followed by filtration using a 5-μm poly-carbonate membrane filter to obtain pure parasites for infection, transfection, and sequencing experiments. Electroporation of the transgenes was performed by using an ECM 830 Square Wave electroporator (BTX). Stable transgenic lines were obtained by selection using 3 μM pyrimethamine or 50 μg/ml xanthine in combination with 25 μg/ml mycophenolic acid, followed by clonal isolation by the limiting dilution method, as previously described[51].

### Primers and plasmids
All primers used in this study were synthesized by Integrated DNA Technologies, and the plasmids were constructed by site-directed mutagenesis of existing plasmids using the Q5® Site-Directed Mutagenesis Kit (New England BioLabs, NEB) or by assembling the DNA fragments using ClonExpress® MultiS One Step Cloning Kit (Vazyme Biotech Co., Ltd, China). All the positive plasmids were verified by DNA sequencing. All primers and plasmids are listed in the Supplementary data 5.

### Endogenous epitope tagging
For C-terminal endogenous tagging, CRISPR-Cas9 plasmids containing a protospacer against the 3′-untranslated region (3′-UTR) of the gene of interest after the stop codon was co-transfected with amplicons flanked with short homology regions including a 2Ty, Ty-AID or 6HA-mAID tag and a drug selection cassette into the respective parasite strain. Successful tagging was verified by PCR and sequencing. For conditional knockdown of the protein of interest, parasites were treated with 500 μM IAA (1:1000 dilution), while mock treatment was performed by addition of 0.1% ethanol only[21].

### Generation of AP2XI-2 knockout
CRISPR-Cas9 mediated homologous gene replacement method was used to construct the AP2XI-2 knockout strain as previously described[51,52]. Briefly, CRISPR-Cas9 plasmids containing a protospacer designed to target the coding region of AP2XI-2 were co-transfected with amplicons containing a DHFR-Ts cassette flanked with ~ 1.5 kb homology arms to the 5′- and 3′- untranslated regions (UTRs) of the AP2XI-2 gene into PruΔ*ku80Δhxgprt* tachyzoites. The successful construction of AP2XI-2 knockouts was confirmed by PCR and DNA sequencing.

### Plaque assay
Fully confluent HFFs grown in 12-well plastic tissue culture plates were infected by 200 tachyzoites of the parental PruΔ*ku80Δhxgprt* or PruΔ*ap2XI-2* strain. For the AID-based strains, the HFFs were infected with an equal number of tachyzoites per well in the presence or absence of 500 μM IAA. After 8-9 days, the cell monolayers were fixed with 4% paraformaldehyde (PFA) for 20 min and stained with 2% crystal violet solution for 30 min at ambient temperature. The number and size of the plaques were determined using a scanner as previously described[50]. All plaque assays were performed in biological triplicate.

### Invasion assay
An immunofluorescence-based invasion assay was performed to assess host cell invasion as previously described[53,54]. Briefly, parasites were grown for 48 h with or without IAA, and intracellular parasites were harvested by scraping the infected host cells and passaging them through a 27-gauge needle. The freshly purified tachyzoites suspended in DMEM medium with or without IAA were added to HFF monolayers grown on coverslips positioned at the bottom of 24 well tissue culture plates and incubated at 37 °C for 30 min. Coverslips were then fixed with 4% PFA and blocked, and the extracellular parasites were stained with primary mouse anti-SAG1 antibody and secondary antibody. The cells were then permeabilized using 0.1% Triton X-100, and all parasites were stained with rabbit anti-GAP45 antibody. The slides were further washed and stained with the respective secondary antibody. Parasites were scored as intracellular or extracellular by using confocal microscopy. Invasion assays were performed in biological triplicate, and at least 15 fields were counted for each replicate.

### Ionophore-induced egress assay
For egress assays, tachyzoites were used to infect cell monolayers in 6-well tissue culture plastic plates for 48-60 h with or without IAA until most of the parasitophorous vacuole (PV) contained approximately a similar number of tachyzoites. Cells were washed three times with prewarmed DMEM and incubated with 3 μM A23187 or control DMSO diluted in DMEM at 37 °C for 3 min. The monolayers were then fixed and permeabilized as described above. The parasites were stained with rabbit anti-GRA5 (PV marker) and mouse anti-IMC1 (whole parasite marker) followed by secondary antibodies to visualize the intact and lysed PVs[54]. The number of the lysed PVs was counted as egress, and at least 100 PVs per plate were examined. The experiment was repeated three independent times.

### Replication assay
Freshly harvested tachyzoites were used to infect HFF monolayers in 12-well tissue culture plastic plates for 2 h. The infected cultures were then washed and incubated with fresh culture medium, with or without IAA. At 30 h post-infection, infected monolayers were fixed and stained with rabbit anti-GAP45, and the number of parasites per PV was determined by using fluorescence microscopy. At least 150 PVs from three biological replicates were assessed to determine the number of tachyzoites per PV.

### In vitro stage differentiation assay
Purified tachyzoites were used to infect HFF monolayers for 4 h under neutral culture conditions (pH 7.4 DMEM containing 2% FBS) with or without IAA. Subsequently, the medium was replaced with alkaline RPMI-HEPES medium containing 2% FBS (pH 8.2) with or without IAA. The cultures were incubated at 37 °C under ambient $CO_2$ levels (~0.03% $CO_2$), and the alkaline medium was replaced daily to maintain a high alkaline pH. The parasite cyst wall was labeled with FITC-conjugated *Dolichos biflorus* lectin (DBL, Vector laboratories), and the parasites

were visualized by staining using different anti-*T. gondii* antibodies. Cyst development was determined by calculating the percentage of DBL-positive PV, and at least 150 PVs were examined from three biological replicates[33,51]. The intensity of DBL staining was categorized into high, middle, or low based on the observed fluorescence intensity.

## Transmission electron microscopy
Infected HFF monolayers maintained under alkaline conditions for 3 days were fixed with 2.5% glutaraldehyde and 0.5% tannic acid in 0.1 M sodium cacodylate buffer at room temperature for 2 h. The samples were then washed with 0.1 M sodium cacodylate buffer and post-fixed in 1% osmium tetroxide and 1.5% potassium ferricyanide for 1 h. After fixation, the samples were dehydrated through a graded series of ethanol and embedded in LX112 resin. Ultrathin sections were cut using an Ultracut UCT (Reichert), stained with uranyl acetate followed by lead citrate, and TEM images were acquired by a Hitachi HT7700 electron microscope at 80 kV[51].

## Immunofluorescence assay
HFF monolayers, grown on glass coverslips placed at the bottom of 12-well tissue culture plastic plates, were infected by *T. gondii*. The infected cells were washed three times with phosphate buffered saline (PBS) and fixed with 4% PFA for 20 min. After five gentle washes with PBS, cells were permeabilized with 0.1% Triton X-100 for 15 min and washed five times with PBS. After blocking in PBS with 5% bovine serum albumin (BSA) at 37 °C for 1 h, cells were incubated with primary antibody at 4 °C overnight or 37 °C for 2 h. Cells were then washed five times with PBS and incubated with secondary antibodies for 2 h. Cell nuclei were counterstained with 4′,6-Diamidino-2-phenylindol (DAPI). After washing five times with PBS, cells were imaged with a Leica confocal microscope system (TCS SP8, Leica, Germany).

## Western blotting
Parasites were harvested from the host feeder cells, syringe filtered, and pelleted by centrifugation at 2000 × *g* for 10 min at 4 °C. The parasite pellets were washed with cold PBS before treating with radioimmunoprecipitation assay (RIPA) lysis buffer containing Protease and Phosphatase Inhibitor Cocktail. The parasite lysates were incubated on ice for 45 min and centrifuged at 12,000 × *g* for 10 min at 4 °C. The supernatants were prepared in Laemmli loading dye, heated at 100 °C for 10 min, and resolved on SDS−polyacrylamide gel electrophoresis (SDS−PAGE). The proteins were then transferred onto nitrocellulose membranes and blocked in 5% fat-free milk-Tris-buffered saline (TBS) supplemented with 0.2% Tween 20 (TBST) and the blots were probed with the primary antibodies, followed by the secondary antibodies. After washing, the membrane was incubated with Pierce enhanced chemiluminescence (ECL) immunoblot substrate to visualize the proteins on a ChemiDoc XRS+ (Bio-Rad, USA).

## Co-immunoprecipitation and mass spectrometry
Immunoprecipitations were conducted by the Pierce Magnetic HA-Tag IP/Co-IP Kit (Thermo Fisher Scientific, USA) according to the manufacturer's instructions. Briefly, Pru::AP2XI-2-mAID-6HA or PruΔ*ku80*Δ*hxgprt* tachyzoites cultured in an alkaline medium for 3 days were collected, syringe filtered, and lysed as described above. The parasites lysate was incubated with mouse monoclonal anti-HA-tag magnetic beads on a rocking platform at 4 °C overnight. After washing, the bound proteins were eluted from the magnetic beads using the elution buffer provided with the kit. The eluted proteins were separated by SDS−PAGE and stained with Coomassie blue. After the excision of eight horizontal gel slices per lane, proteins were in-gel digested with trypsin. The peptides separated by liquid phase chromatography were ionized by a nanoESI source for tandem mass spectrometry (LC-MS/MS) analysis using the Q-Exactive HF-X mass spectrometer (Thermo Fisher Scientific, San Jose, CA, USA) for DDA

(Data Dependent Acquisition) mode detection. The MS data was searched using the Mascot tool against the protein database containing *T. gondii* ME49-translated open reading frames in the ToxoDB (https://toxodb.org)[55]. The output was filtered with a false discovery rate (FDR) 1% at spectral level (PSM level FDR ≤ 0.01).

## RNA sequencing
We performed RNA-Seq on tachyzoites of AP2XI-2-mAID and AP2XII-1-mAID strains cultured under neutral or alkaline conditions with or without IAA. Briefly, tachyzoites were allowed to infect HFF monolayers for 4 h under neutral conditions with or without IAA. Then, infected monolayers were washed with DMEM, and the medium was replaced with neutral medium or alkaline medium, with or without IAA. After 72 h post-infection, the parasites were isolated from the host cells as described above, washed, and the parasite pellets were used to extract RNA using the RNeasy kit (Qiagen, MD, USA). All extracted RNA samples were treated with RNase-Free DNase to remove residual genomic DNA. The integrity and quantity of the isolated RNA were measured using a Nano Drop and an Agilent 2100 bioanalyzer, respectively. Sequencing libraries were generated, amplified, and subjected to 150-bp paired-end sequencing using an MGISEQ 2000 (BGI-Shenzhen, China). The raw data were processed to remove reads containing sequencing adapters, low-quality reads, and reads with ≥ 5% unknown base ('N' base) using SOAPnuke[56]. The clean reads were aligned against the *T. gondii* ME49 reference genome (https://toxodb.org) using Bowtie2[57] and the reads per kilobase per million mapped reads (RPKM) method was employed to calculate the relative gene expression with RSEM[58]. DESeq2 software was used to determine gene expression and identify the differentially expressed genes (DEGs)[59]. Genes with a $|\log_2$ fold change (FC)$| \geq 1$ and a *p*-value of <0.05 were deemed significantly differentially expressed. Volcano plots and heatmaps were constructed using R (https://www.r-project.org/) and TBtools[60], respectively, to visualize the gene expression differences between different strains.

## Cleavage under targets and tagmentation (CUT&Tag) assay
The CUT&Tag assay was performed as previously described with modifications using the Hyperactive Universal CUT&Tag Assay Kit for Illumina (Vazyme Biotech, #TD903, China)[61]. Briefly, ~ 10^7 intracellular tachyzoites were collected, and the parasite nuclei were isolated by resuspending the washed parasite pellet in NE Buffer. The nuclei were then processed according to the manufacturer's protocol, and libraries were constructed using 'TD903 Hyperactive Universal CUT&Tag Assay Kit for Illumina' and 'TD202 TruePrep Index Kit V2 for Illumina' from Vazyme Biotech (#TD202, China). The libraries were sequenced on Illumina Novaseq platform at Novogene Science and Technology Co., Ltd (Beijing, China), producing 150 bp paired-end reads. Raw reads in fastq format were processed using fastp (v0.20.0)[62] to obtain clean reads, which involved removing reads containing adapter, reads containing ploy-N and low-quality reads from the raw data. Clean reads were mapped against the ME49 genome using BWA (v0.7.12)[63] and only uniquely mapped (MAPQ ≥ 13) and de-duplicated reads were used for further analysis. Peak calling was performed with MACS2 (v2.1.0)[64] and ChIPseeker[65] was used to retrieve the nearest genes around the peak and annotate the genomic region of the peak. Visualization of peak distribution along genomic regions of the genes of interest was performed with IGV[66]. ChIP-seq of MORC-HA Rep1 and MORC-HA Rep2 were obtained from the GEO under accession number GSE136060[14].

## Virulence assays in mice
Eight-week-old female C57BL/6 mice, obtained from the Center of Laboratory Animals of Lanzhou Veterinary Research Institute, were maintained under pathogen-free and controlled conditions (12/12−h dark/light cycle, 50−60% humidity, and 22 °C temperature), with free access to sterilized food and water. After 1-week of acclimatization, the

mice were infected intraperitoneally (i.p.) by $2\times10^4$ or $2\times10^5$ freshly egressed tachyzoites of the parental or PruΔ*ap2XI-2* strain. The mice were closely monitored for morbidity and observed for humane endpoint twice a day for up to 30 days. Mice that reached the humane endpoint were immediately euthanized. Brain cyst burden was determined in the survived mice at 30 days post-infection. In brief, brains were collected in PBS, homogenized, and the number of the brain cysts was determined by using FITC-conjugated DBL staining[67]. Parasite cyst burden was determined based on the number of cysts detected in at least one cerebral hemisphere per mouse brain. Housing of mice and all procedures involving animal experimentation were conducted in strict accordance with institutional guidelines approved by the Animal Research Ethics Committee of Lanzhou Veterinary Research Institute, Chinese Academy of Agricultural Sciences (LVRI-2020-18).

## Statistics and reproducibility

All statistical analyses were performed using Prism software version 7.01 (GraphPad Software Inc., CA, USA). Tables were visualized with Microsoft Word or Excel. Data are expressed as the mean ± standard deviation (SD). Information including the number of observations, the number of biological replicates, error bars, and the exact statistical tests used, can be found in the figure legends. Statistical significance was determined using a threshold of $p < 0.05$. All microscopy images are representatives of at least two independent experiments, and all experiments resulted in comparable results.

## Data availability

The RNA-Seq data have been deposited in the short read archive (SRA) of the NCBI under the accession number PRJNA990465. The CUT&Tag data have been deposited in the GEO under accession number GSE248448. The mass spectrometry proteomics data of AP2XI-2 interactome have been deposited in the ProteomeXchange Consortium via the PRIDE partner repository with the dataset identifier PXD043808. *Toxoplasma gondii* genome information was obtained from the ToxoDB (https://toxodb.org) and Eukaryotic pathogen, Vector & Host Informatics Resources were obtained from the VEupathDB (https://veupathdb.org). Source data are provided with this paper.

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

## Acknowledgements

We thank Prof L. David Sibley at the Department of Molecular Microbiology, Washington University School of Medicine in St. Louis for valuable comments and constructive suggestions on the draft manuscript. The authors acknowledge the technical assistance of BGI-Shenzhen in RNA-Seq and proteomics analyses and Novogene in CUT&Tag analysis. This work was supported by the National Key Research and Development Program of China (Grant Nos. 2021YFC2300800, 2021YFC2300802 and 2022YFD1800200) to X.Q.Z. and J.L.W., the National Natural Science Foundation of China (Grant No. 32172887) to X.Q.Z., the Natural Science Foundation of Gansu Province, China (Grant Nos. 23JRRA555 and 23JRRA1479) to J.L.W., the Research Funding from the Lanzhou Veterinary Research Institute, Chinese Academy of Agricultural Sciences (Grant No. CAAS-ASTIP-JBGS-20210801) to J.L.W., and the Special Research Fund of Shanxi Agricultural University for High-level Talents (Grant No. 2021XG001) to X.Q.Z. The funders had no role in the study design, data analysis, data interpretation, and the writing of this report. All authors had full access to the data in the study and accept the responsibility to submit it for publication.

## Author contributions

J.L.W., X.Q.Z., and H.M.E. conceived and designed the study. J.L.W., T.T.L., and N.Z.Z. performed the experiments and analyzed the data

with inputs from M.W., L.X.S., and Z.W.Z. J.L.W. and T.T.L. wrote the manuscript and produced the figures. B.Q.F., H.M.E., J.L.W., and X.Q.Z. critically revised and edited the manuscript. X.Q.Z. and J.L.W. secured the funds. X.Q.Z. and J.L.W. supervised the project. All authors reviewed and approved the final version of the manuscript.

## Competing interests

The authors declare no competing interests.
