## [Peer Review File · Nature Communications]

REVIEWER COMMENTS

Reviewer #1 (Remarks to the Author):

The manuscript entitled "The transcription factor AP2XI-2 is a key negative regulator of *Toxoplasma gondii* merogony" describes studies of two ApiAP2 factors (AP2XI-2 and AP2XII-1) expressed in the tachyzoite and bradyzoite stages of the intermediate life cycle. These AP2 factors were previously identified as interactors of the MORC complex that is responsible for repressing definitive life cycle stage (sexual stages) gene expression in the intermediate stages. The studies appear to validate a role for AP2XI-2 and AP2XII-1 in suppressing some merozoite stage gene expression. Conditional knockdown of AP2XI-2 combined with an unknown mechanism induced by exposing parasites to alkaline-stress also leads to abnormal multinuclear reduplication. Knockdown of AP2XII-1 also has a similar gene expression phenotype and causes nuclear reduplication. Based on these results, the authors conclude AP2XI-2 and AP2XII-1 are major regulators of merozoite replication, which occurs by endopolygony. They further conclude that deletion of these factors causes tachyzoites to bypass the bradyzoite stage and switch directly to the merozoite. Whether these conclusions are justified is offset by major flaws in the study. The authors have not considered alternative cell cycle interpretations of their results and the use of a developmentally compromised parent strain greatly complicates developmental conclusions. The results as presented often lack the precision and depth that is needed to support their conclusions. Overall, this is an interesting study that may have been prematurely submitted.

Specific comments:

Figure 1 and S1

--Using the acronym Ctrl for the parent strain and the IAA- condition is very confusing. Please uniquely differentiate these strains in the figures. The western blot of AP2XI-2 has multiple bands that are lost with IAA treatment. There is a single mass standard indicated, so which band is AP2XI-2 (not indicated), and why are there multiple bands.

--Given the importance of an unknown mechanism induced by alkaline media, it is critical that this condition be comprehensively evaluated. In our hands, days of parasite exposure to alkaline stress kills high numbers of parasites (80% death in some measurements). In this Figure 1, alkaline-stress was largely not evaluated in the parent or transgenic replication, invasion, or egress. In addition, viability in alkaline media was not addressed, which can be quantified by plaque number (not plaque size), although there are other ways to approach this question.

--It is well understood that forced adaptation and repeated passage in cell culture leads to severe losses in *Toxoplasma* developmental competency. The highly modified Pru strain used here has been passed hundreds if not thousands of times in cell culture. Consequently, it is not surprising that the Pru parent used here produces a couple of dozen cysts in the brain of mice. These numbers should have been a warning that the strain is compromised. Native strains, including the original Pru strain, produce many hundreds to thousands of tissue cysts in mouse brain. It is unlikely that the strain used here could complete the cat life cycle, which is critical to the subject of this paper. Note that in the recent discovery of the BFD1 master regulator, the Lourido lab was careful to confirm their findings in a developmentally competent strain, which should have been done here as well.

Figures 3 and S2 gene expression

--In these figures and throughout the paper, heat maps of gene expression based on fold change are presented. The authors do not provide any fold change data in the supplement documents nor is there any statistical validation of the RNA-seq data included. It is unreasonable to require readers to regenerate fold change data from the FPKM results, which is the only data included in the supplement.

--Heat maps of selected gene sets could be interpreted as "cherry picking". In the methods, the authors indicated they were including volcano plots (pg 23, line 625) which show comprehensive changes in mRNA expression versus statistical significance on a larger scale. Where are those plots?

--The authors indicate AP2XI-2 knockdown combined with alkaline-stress switches the tachyzoite to merozoites but mostly show developmental SRS gene expression. The published analysis of native merozoite (from cats) gene expression discovered >300 genes uniquely expressed >8-fold in merozoites. It is not clear the authors used this gene set in their study. A quick look at the 1000 highest expressed mRNAs in the alkaline/IAA+ XI-2aid parasites from their supplemental data only hit 15% of the 312 top native merozoite mRNAs. It is troubling that AP2VIIa-1, which is the only AP2 factor highly expressed in native merozoites (>10-fold) was not significantly induced in the XI-2aid parasites. Clearly, a more comprehensive analysis of known merozoite gene expression in this study is needed.

--One of the other findings of these figures was the lack of overlap between the mRNAs induced by alkaline/IAA+ in XI-2aid parasites and bradyzoite mRNAs, which led to the authors to conclude that tachyzoites are switching directly to merozoites. However, the reliance on in vitro/alkaline datasets is a serious flaw. Alkaline-stress only induces early bradyzoite gene expression. The merozoite genes selected for the heat maps in Fig 3 and S2 substantial overlap with mRNAs elevated in native, mature bradyzoites (databases that are available from ToxoDB); a third of the merozoite genes in Fig 3a and half the family A,B,C,D,E genes in Fig S2a are expressed (some far higher than 10-fold) in in vivo bradyzoites. The authors need to revisit their analysis of bradyzoite gene expression using more appropriate datasets. It is possible that using a developmentally competent strain would strengthen their study.

Figures 2, 3, 4 and supplement cell biology data

--The paper relies primarily on IFA images from cultures infected at high MOI and exposed to alkaline-media and IAA for 3 days. Many host cells are multiply infected, and the vacuoles are large and chaotic due to the long accumulation of abnormalities, which makes it challenging to understand the phenotypes. Keep in mind that *Toxoplasma* can complete chromosome replication and nuclear division in 4 h or less, so even 24 h is often too long to carefully follow each nuclear division let alone 3 days. The authors provide multiple examples of IFA images as a substitute for quantification in this figure and throughout the paper. It is critical to see images of the phenotype developing (more early timepoints) and to show quantitation of the key phenotype characteristics (e.g. abnormal vacuole numbers, nuclei counts over time, merozoite yields ect.). A good example is the clear evidence that the AP2XII-1aid is undergoing endodyogeny and endopolygeny in IAA+ conditions (see Fig. 5 Day 1 images, the lower left corner parasites are dividing by endodyogeny). The fraction of binary vs >binary division needs to be quantified throughout this study. Bottom line, we need to see images and quantification of single parasite infections from the first division onwards.

--The free parasites assumed to be merozoites need to be used in a second infection of host cells to determine whether the multinuclear phenotype is stable, or does the parasite revert to binary division (under neutral pH). Perhaps what is thought to be free merozoites are residual tachyzoites, which a secondary infection could help sort out (replication will be binary). Do these merozoites differentiate into sexual stages, paying attention to whether the host cell type needs to be changed or utilize the new linolenic acid models to test the function of these merozoites. These experiments are critical to determine whether these parasites are a tachyzoite cell cycle mutant with merozoite genes derepressed or true functional merozoite. The authors show they can make direct knockouts of either AP2 factor, so the IAA/aid model is not really needed.

--In evaluating cyst wall formation in this figure and other figures, the authors present a high, middle, low criteria for DBA staining, yet they provide no image references for these criteria.

--What we know about tachyzoite replication does not appear to be considered in this paper. Firstly, tachyzoites replicate by nuclear counts greater than binary in 1-3% in cell culture and this might be higher in alkaline media (needs to be evaluated). Secondly, we know that the centrosome of tachyzoites is uniquely bipartite with separate complexes controlling karyokinesis versus cytokinesis (budding). It is very easy to uncouple these mechanisms, which often leads to multinuclear replication. Abnormal multinuclear replication is one of the most common tachyzoite cell cycle mutant phenotypes. The images presented in this paper of large-scale nuclear reduplication are in most cases severely abnormal and are likely not viable--similar to cell cycle mutants. If the parasites are viable, the authors need to show that data. The centrin staining in this paper is unconvincing, and more importantly, the centrin complex controls budding not karyokinesis. The authors are using the wrong marker to follow the complex that controls nuclear replication (see Suvorova et al PloS Biol).

--The authors interpretation of IMC7 might need a reexamination. The inner membrane complex of daughter parasites is derived from both the mother and new synthesis, and thus, mature merozoites are positive for IMC7 from the mother. It is only the late schizont prior to budding completion that is IMC7 negative (look again at reference 27). More importantly, IMC7 is not just negative or positive here. IMC7 is extensively staining abnormal membranes (see Fig. 3c) so all bets are off about what this means developmentally, it is certainly not native IMC7 patterns.

--the authors had the right idea in using TEM to analyze parasite replication at higher resolution. However, three poor images with none showing daughter formation in their mutants falls far short of what was needed.

--Where are the IAA- staining images for tubulin and ARO protein in Fig. S2b?

--It is not clear there is any value of the BFD1 knockout or knocking out XI-2 in RH experiments. Both the Pru parent and RH parent are developmentally compromised strains. The staining of XI-2-HA in the RH strain is non-nuclear and completely abnormal with or without IAA.

Figure 7--other AP2 factors

--The study of other AP2 factors in this paper contributes little to this paper and could be artifact on top of artifact. A more comprehensive study of XI-2 and XII-1 that includes evaluating the function of these two AP2s in developmentally competent strains is a better use of time and energy.

Reviewer #2 (Remarks to the Author):

The ultimate product of *Toxoplasma gondii* sexual development (the sporulated oocyst) accounts for roughly 50% of human infection. However, little is known at the molecular level about *Toxoplasma* presexual development because this process is restricted to the cat gut and in vitro systems to study this stage are lacking. This manuscript details the role of two AP2 family transcription factors (AP2XI-2 & AP2XII-1) in preventing parasite sexual commitment by directing the HDAC3/MORC complex to transcriptionally silence gene expression. The work presented here closely mirrors a recent preprint (posted 01/2023, PMID: 36711883) by the Hakimi group. There is substantial overlap between this manuscript and the Hakimi preprint. Both works reach the same major conclusions: 1) AP2XI-2 & AP2XII-1 work in a complex to direct the HDAC3/MORC complex to transcriptionally silence gene expression, & 2) AP2XI-2 and/or AP2XII-1 knockdown drives merozoite formation. Whereas the Hakimi work is more developed on the genomics & proteomics side, the work reviewed here slightly extends beyond the Hakimi preprint by assessing the role of AP2XI-2/XII-1 accessory proteins to merozoite formation and validating the downstream expression of additional AP2 transcription factors that have roles yet to be defined. The manuscript tells a cohesive, complete, and intriguing story that would be of interest to broad readership interested in gene regulation and/or parasitology.

- Fig 1H: the cyst burden is atypically low. What is the limit of detection in this assay? Was a PCR performed to measure parasite burden in the brain? Was serology performed to ensure infection in all mice? The methods section should state what the inoculum was in this section (not just the text body and figure legend). Data should be presented about weight loss over the time of acute infection.

- RNAseq analysis: states that bowtie2 was used for alignment which is atypical because it doesn't allow for spliced mapping. More importantly, the methods also state that differential gene expression analysis was performed with DESeq2. This data was not provided in supplement (should have log2FC and stats associated with the call for each gene, not FPKM as provided). There are no volcano plots in the manuscript as presented (as mentioned in the methods) – just heatmaps for selected genes. It would be informative to show volcano plots though because it would allow the reader to assess to what degree genes are (dys)regulated under each condition. Preferably, genes presented in a volcano plot format could still be color-coded based on their canonical stage-specific expression. Sporadically throughout the manuscript, the RNAseq data is stated to show changes in protein expression as opposed to transcript abundance. Example: lines 300-1.

- The deficiency in bradyzoite formation is detailed in Fig 2B & Fig 4B. What proportion of parasites are undergoing the merogony phenotype? What about in the deltaBFD1 line?

- Line 228-9: please substantiate the slow growth claim by assessing whether alkaline media does in fact reduce the growth rate independent of BFD1 expression in type your I/II parasite lines. (stated in ref 30 – lourido BFD1 paper)

- To what degree do AP2XII1/AP2XI-2 and MORC peaks overlap? What proportion of MORC-resident genes are XII-1/XI-2 dependent? Does MORC occupancy decrease at these genes upon XII1/XI2 depletion?

- Line 310-2: if both AP2XII-1 and XI-2 bind each other (co-IP experiment), probably acting as a heterodimer, what model is proposed where one works upstream of another? Are there peaks that are

uniquely APXII-1 or AP2XII-2 from the CUT&Tag to support a model where these factors could work independently? For example could they form a mixture of homo- and heterodimers that could explain their differences?

- Fig 5A: the last column shows egressed merozoites. Can they reinvade? And what happens to do them under the second growth cycle and beyond?

- Fig 6: since statements are given in the text about the relative switching efficiency under neutral and alkaline culture conditions, please categorize and quantify in the figure. This is relevant because the statistical term “significantly increased” is used on line 307. Without this information, it is hard to ascertain to what degree the culture conditions matter in the double knockdown. Same comment applies throughout the paragraphs from lines 306-323 since quantitative comparisons are being made.

- Fig 7A: are each of these AP2s turned on only by XI-2 knockdown? Please show IFA to see whether they are turned in under normal neutral or alkaline (without IAA) conditions. Also, which (if any) of these have been shown to complex with MORC – would give indication about whether they are transcriptional activators or repressors.

MINOR:

- Line 122: Suggest removing the term “two-color assay” assay since it is not mentioned by that name in the Materials and Methods section and the term is not informative to a general readership audience. Alternatively, modify M&M accordingly.

- Lines 141-52: given the generalist readership of this journal, explain the importance of the IMC1 staining as marking the mother in this paragraph and how that contributes to the conclusion that the parasites have ‘multinucleated morphology’ as mentioned in the text.

- Fig S2D: what is the red stain for this panel? If IMC1, it would be helpful to recolor the figure to mirror S2C

- Lines 167-8: soften the language about these genes being tachyzoite-specific (SAG1 is tachyzoite specific; GRA1 is in bradyzoites PMID: 18006371; MIC2 is lowered but not gone in bradyzoites PMID: 34860156; not sure about ROP1). What about other canonical tachyzoite markers (like ENO2)? Suggest to reword saying that their transcripts have their peak expression in tachyzoites if this is the case.

- Fig S2B typo (IMC17A->MIC17A)

- Fig S2C: please provide labels showing key features described in the text on the micrographs.

- Fig S2D: what is the red stain for this panel?

- Line 211: typo “was no observed”

- Fig S3F: panels are mislabeled (HA / DNA).

- Line 254: typo were->was

- Line 291: typo nucleuses->nuclei

- Fig 6: each panel is very small and hard to see what is going on without really zooming in.

- Fig 7A: suggest adding a label to the figure to indicate the categories mentioned in the text to allow for quick referral (e.g. IV-3, IV-2, VIIa-1 are present in mature merozoites).
- Line 388: change “unable to regulate bradyzoite genes” to “unable to induce bradyzoite genes”
- Line 437: typo w,hereas
- Line 600: what is the elution buffer?
- Reference used for the MORC ChIPseq data should be provided in the appropriate section of the methods and in the body of the text.

Reviewer #3 (Remarks to the Author):

In this manuscript, Wang and colleagues identify two putative transcription factors from the eukaryotic pathogen *Toxoplasma gondii* that play a critical role in regulating stage-specific gene expression in the parasite. They demonstrate that AP2XII-1 and AP2XI-2 are expressed in the tachyzoite and bradyzoite stages and repress genes that are normally expressed during merogony and in the mature merozoite. Knockdown of either (or both) of these regulators results in upregulation of merozoite-specific genes with concurrent downregulation of tachyzoite genes and is associated with changes in parasite morphology that is consistent with merogony and mature merozoites. This is further supported with a large number of inducible knockdown lines of genes encoding proteins that interact with AP2XII-1 and AP2XI-2. Importantly, this work demonstrates the feasibility of in vitro production and study of *Toxoplasma* merozoites and will be of great interest to the Apicomplexan field.

The study is complementary to a recent preprint from Antunes et al (<https://doi.org/10.1101/2023.01.16.524187>) and the conclusions from both groups are aligned. Although there is a degree of overlap between the central hypotheses of the two manuscripts, an important distinction is that Wang et al have performed their analyses in a Type II Pru parasite background (compared to the Type I RH background in Antunes et al). Significantly, Wang et al describe AP2 knockdown in the context of alkaline stress which appears to be a strong trigger of merogony in the absence of either XI-2 or XII-1.

The experiments have been performed rigorously with sufficiently detailed methods in most cases.

Minor comments and suggestions:

The change in morphology to “significantly thinner” or “multinucleated” is used as a marker for merogony and merozoite formation throughout the paper, but all the images that are presented are zoomed out, and in some cases (eg Fig 2a) the DAPI channel is unfocused and blurry, making it difficult to clearly define these different morphologies. It would be helpful to include a few enlarged inset images of just a few parasites to clearly demonstrate these two morphologies and allow the reader to compare to previously published images of enteroepithelial stages.

Most of the figures containing imaging data include 3 or more examples of the different morphologies that were observed (figs. 2a, 3c, 5 and 6). The authors might consider streamlining this data a little to help the reader, and include just one example each of the “thinner” and “multinucleated” morphologies.

Furthermore, annotating the images with arrows or labels of putative schizonts/merozoites would help with reader comprehension.

How were “bradyzoite-specific” and “merozoite-specific” genes defined in Figure 2 and 3? What were the criteria used to select these lists of genes?

Figure 2b. How was “high, middle and low” DBL staining defined? This should be explained in the methods.

Line 291: nucleuses should be nuclei

Line 844, Fig 1c legend: (c) should be (b)

26 November, 2023

Dear Reviewers,

Re: Revised manuscript NCOMMS-23-30669.R1

On behalf of all co-authors, I would like to thank the three reviewers very much for the positive and favorable comments, and constructive suggestions on our manuscript (MS) NCOMMS-23-30669. The three reviewers had a positive view on our MS and considered that the data is of significant and broad interest to the readership of the esteemed journal *Nature Communications*. We are especially grateful to the reviewers for their efforts in taking the time to provide detailed and constructive comments and thoughtful suggestions which we believe have improved the quality of our MS.

We apologize that the revision has taken a bit longer than expected because we have performed additional experiments suggested by the reviewers and have taken efforts to ensure full and thorough addressing of the reviewers' comments and suggestions. We have revised the MS strictly according to the reviewers' comments and suggestions, and we have done the requested additional experiments and all the required clarifications. Two MS files are uploaded: one shows the text with "**tracked changes**", and the other with the unannotated text is a "**clean file**". In the following, we detail our point-by-point responses to the reviewers' comments and suggestions.

Responses to comments and suggestions of Reviewer #1

General comments:

The manuscript entitled "The transcription factor AP2XI-2 is a key negative regulator of *Toxoplasma gondii* merogony" describes studies of two ApiAP2 factors (AP2XI-2 and AP2XII-1) expressed in the tachyzoite and bradyzoite stages of the intermediate life cycle. These AP2 factors were previously identified as interactors of the MORC complex that is responsible for repressing definitive life cycle stage (sexual stages) gene expression in the intermediate stages. The studies appear to validate a role for AP2XI-2 and AP2XII-1 in suppressing some merozoite stage gene expression. Conditional knockdown of AP2XI-2 combined with an unknown mechanism induced by exposing parasites to alkaline-stress also leads to abnormal multinuclear reduplication. Knockdown of AP2XII-1 also has a similar gene expression phenotype and causes nuclear reduplication. Based on these results, the authors conclude AP2XI-2 and AP2XII-1 are major regulators of merozoite replication, which occurs by endopolygony. They further conclude that deletion of these factors causes tachyzoites to bypass the bradyzoite stage and switch directly to the merozoite. Whether these conclusions are justified is offset by major flaws in the study. The authors have not considered

alternative cell cycle interpretations of their results and the use of a developmentally compromised parent strain greatly complicates developmental conclusions. The results as presented often lack the precision and depth that is needed to support their conclusions. Overall, this is an interesting study that may have been prematurely submitted.

Response: We thank the reviewer very much for the favorable comments and constructive suggestions on our MS. We have conducted additional experiments to demonstrate that the mutant parasites are indeed functional merozoites and not merely tachyzoite cell cycle mutants. Additionally, up to 95% of the parental strains used in our study can form cysts *in vitro*. The low number of cysts observed *in vivo* is caused by the rapid loss of developmental competency of parasites cultured *in vitro*, even in the ME49EW strain, which has been exclusively maintained *in vivo* (Ref, PMID: 37675999, DOI: 10.1128/mbio.01836-23). As suggested by the editors, we have clarified this point in the discussion.

Specific comments:

Comment: Figure 1 and S1

--Using the acronym Ctrl for the parent strain and the IAA- condition is very confusing. Please uniquely differentiate these strains in the figures. The western blot of AP2XI-2 has multiple bands that are lost with IAA treatment. There is a single mass standard indicated, so which band is AP2XI-2 (not indicated), and why are there multiple bands.

Response: We thank the reviewer very much for the suggestion. We have replaced the acronym 'Ctrl' with '-IAA' to avoid confusion. In the Western blot analysis, we have indicated the AP2XI-2 band. The presence of multiple bands may reflect instability of the parasite lysate or due to posttranscriptional modifications. We have revised the figure legends accordingly.

Comment: Given the importance of an unknown mechanism induced by alkaline media, it is critical that this condition be comprehensively evaluated. In our hands, days of parasite exposure to alkaline stress kills high numbers of parasites (80% death in some measurements). In this Figure 1, alkaline-stress was largely not evaluated in the parent or transgenic replication, invasion, or egress. In addition, viability in alkaline media was not addressed, which can be quantified by plaque number (not plaque size), although there are other ways to approach this question.

Response: We thank the reviewer very much for the suggestion. The most commonly used method to induce bradyzoite cysts in *in vitro* culture is through exposure to continuous alkaline culture condition. When employing this method, we observed that up to 95% of the parental Pru strain can form bradyzoites, and these bradyzoites were viable. In the alkaline medium, AP2XI-2-depleted parasites typically undergo merogony, usually forming one or two meronts per parasitophorous vacuole, 24 hours post-infection. In contrast, parasites expressing AP2XI-2 develop into bradyzoites. *Toxoplasma* invasion is severely impaired in an alkaline medium, even in the parental Pru strain. Moreover, it may not be suitable to evaluate

parasite egress under alkaline conditions due to the presence of two different parasite stages. The induction of merogony by alkaline medium may be attributed to the slower growth of the parasites or the expression of other factor(s) induced by alkaline medium.

Comment: It is well understood that forced adaptation and repeated passage in cell culture leads to severe losses in *Toxoplasma* developmental competency. The highly modified Pru strain used here has been passed hundreds if not thousands of times in cell culture. Consequently, it is not surprising that the Pru parent used here produces a couple of dozen cysts in the brain of mice. These numbers should have been a warning that the strain is compromised. Native strains, including the original Pru strain, produce many hundreds to thousands of tissue cysts in mouse brain. It is unlikely that the strain used here could complete the cat life cycle, which is critical to the subject of this paper. Note that in the recent discovery of the BFD1 master regulator, the Lourido lab was careful to confirm their findings in a developmentally competent strain, which should have been done here as well.

Response: We thank the reviewer for the constructive suggestion. Recently, the Michael W. White lab reported that the adaptation of ME49EW parasites in fibroblasts, which have been exclusively maintained *in vivo*, can lead to a permanent loss of cyst development in mice. ME49EW tachyzoites also produce a few dozen cysts in the brains of mice, suggesting that adaptation to cell culture, rather than repeated passage, is the primary reason for the low number of cysts *in vivo* (Ref, PMID: 37675999, DOI: 10.1128/mbio.01836-23). Additionally, up to 95% of the parental Pru strains used in our study formed cysts *in vitro*, suggesting that the ability of the parental Pru strains to complete the life cycle, at least to form bradyzoites, was not compromised. As suggested by the editors, we have highlighted this important point in the discussion.

Comment: Figures 3 and S2 gene expression

--In these figures and throughout the paper, heat maps of gene expression based on fold change are presented. The authors do not provide any fold change data in the supplement documents nor is there any statistical validation of the RNA-seq data included. It is unreasonable to require readers to regenerate fold change data from the FPKM results, which is the only data included in the supplement. Heat maps of selected gene sets could be interpreted as "cherry picking". In the methods, the authors indicated they were including volcano plots (pg 23, line 625) which show comprehensive changes in mRNA expression versus statistical significance on a larger scale. Where are those plots?

Response: We greatly appreciate the reviewer for the constructive suggestions, and we have incorporated comprehensive changes in mRNA expression into the volcano plots, with genes presented in a color-coded format based on stage-specific expression. In addition, the log₂ fold change data are provided in the Supplementary Dataset 1.

Comment: The authors indicate AP2XI-2 knockdown combined with alkaline-stress switches

the tachyzoite to merozoites but mostly show developmental SRS gene expression. The published analysis of native merozoite (from cats) gene expression discovered >300 genes uniquely expressed >8-fold in merozoites. It is not clear the authors used this gene set in their study. A quick look at the 1000 highest expressed mRNAs in the alkaline/IAA+ XI-2aid parasites from their supplemental data only hit 15% of the 312 top native merozoite mRNAs. It is troubling that AP2VIIa-1, which is the only AP2 factor highly expressed in native merozoites (>10-fold) was not significantly induced in the XI-2aid parasites. Clearly, a more comprehensive analysis of known merozoite gene expression in this study is needed.

Response: We thank the reviewer very much for the constructive suggestion, and we have conducted a more comprehensive RNA-seq analysis. Comparative RNA-seq analysis revealed that approximately 49% of the merozoite-specific transcripts (152 out of 312) were significantly induced (>2-fold) after depletion of AP2XI-2 when cultured under alkaline conditions. The induction of only 49% of the merozoite-specific transcripts may be attributed to the fact that most AP2XI-2-depleted parasites are schizonts and do not form mature merozoites, with only 10-15% forming mature merozoites after 3 days of growth under alkaline conditions. In addition, the expression level of AP2VIIa-1 (TGME49_280470) was upregulated (~ 2-fold) and confirmed by IFA in the Figure 7, showing that AP2VIIa-1 is only expressed in mature merozoites.

Comment: One of the other findings of these figures was the lack of overlap between the mRNAs induced by alkaline/IAA+ in XI-2aid parasites and bradyzoite mRNAs, which led to the authors to conclude that tachyzoites are switching directly to merozoites. However, the reliance on in vitro/alkaline datasets is a serious flaw. Alkaline-stress only induces early bradyzoite gene expression. The merozoite genes selected for the heat maps in Fig 3 and S2 substantial overlap with mRNAs elevated in native, mature bradyzoites (databases that are available from ToxoDB); a third of the merozoite genes in Fig 3a and half the family A,B,C,D,E genes in Fig S2a are expressed (some far higher than 10-fold) in in vivo bradyzoites. The authors need to revisit their analysis of bradyzoite gene expression using more appropriate datasets. It is possible that using a developmentally competent strain would strengthen their study.

Response: We thank the reviewer very much for the constructive suggestion. Comparative RNA-seq analysis using existing datasets generated from early and mature bradyzoites showed that only approximately 5.8% of the bradyzoite-specific transcripts (17 out of 291) were significantly abundant (>2-fold) in the AP2XI-2-depleted parasites, while 33% of the bradyzoite-specific transcripts (97 out of 291), including the classic bradyzoites markers BAG1, CST1 and LDH2 were significantly decreased in the AP2XI-2-depleted parasites when cultured under alkaline conditions. Additionally, in the previous Fig 3a and Fig S2a, the majority of (108/117) merozoite genes were not expressed in bradyzoites *in vivo*. We have added the bradyzoite-specific genes in the volcano plots and found that most classic bradyzoite-specific genes were not activated in the AP2XI-2-depleted parasites.

Comment: Figures 2, 3, 4 and supplement cell biology data

--The paper relies primarily on IFA images from cultures infected at high MOI and exposed to alkaline-media and IAA for 3 days. Many host cells are multiply infected, and the vacuoles are large and chaotic due to the long accumulation of abnormalities, which makes it challenging to understand the phenotypes. Keep in mind that *Toxoplasma* can complete chromosome replication and nuclear division in 4 h or less, so even 24 h is often too long to carefully follow each nuclear division let alone 3 days. The authors provide multiple examples of IFA images as a substitute for quantification in this figure and throughout the paper. It is critical to see images of the phenotype developing (more early timepoints) and to show quantitation of the key phenotype characteristics (e.g. abnormal vacuole numbers, nuclei counts over time, merozoite yields ect.). A good example is the clear evidence that the AP2XII-1aid is undergoing endodyogeny and endopolygeny in IAA+ conditions (see Fig. 5 Day 1 images, the lower left corner parasites are dividing by endodyogeny). The fraction of binary vs >binary division needs to be quantified throughout this study. Bottom line, we need to see images and quantification of single parasite infections from the first division onwards.

Response: We thank the reviewer very much for the constructive suggestion. As shown in Figure 5a and Supplementary figures 5, more than 95% of the AP2XI-2-depleted parasites undergo merogony, typically forming one or two schizonts per parasitophorous vacuole after 24 hours post-infection in the alkaline medium. This process involves nuclear endodyogeny division. We have added the suggested data in the revised Figure 6c and Supplementary Figure 6.

Comment: The free parasites assumed to be merozoites need to be used in a second infection of host cells to determine whether the multinuclear phenotype is stable, or does the parasite revert to binary division (under neutral pH). Perhaps what is thought to be free merozoites are residual tachyzoites, which a secondary infection could help sort out (replication will be binary). Do these merozoites differentiate into sexual stages, paying attention to whether the host cell type needs to be changed or utilize the new linolenic acid models to test the function of these merozoites. These experiments are critical to determine whether these parasites are a tachyzoite cell cycle mutant with merozoite genes derepressed or true functional merozoite. The authors show they can make direct knockouts of either AP2 factor, so the IAA/aid model is not really needed.

Response: We thank the reviewer very much for the constructive suggestion. We have done the experiment suggested and after the second cycle of infection of HFFs, mature merozoites did not invade the host cell under normal medium conditions. However, schizonts differentiated into tachyzoites when the medium is changed to a neutral one without IAA. AP2XI-2 can be directly knocked out, whereas AP2XII-1 cannot be directly knocked out. However, some experiments could not be easily conducted using the AP2XI-2 knockouts due to their slow growth. Thus, most experiments were conducted by using the AP2XI-2 depleted

parasites. We thank the reviewer very much for the excellent suggestions on the need to use cat epithelial cells and/or the linolenic acid models to investigate the function of these merozoites in our further studies.

Comment: In evaluating cyst wall formation in this figure and other figures, the authors present a high, middle, low criteria for DBA staining, yet they provide no image references for these criteria.

Response: We have explained this stratification of the DBL staining in the methods. We have also added image references for the DBL staining criteria as requested.

Comment: What we know about tachyzoite replication does not appear to be considered in this paper. Firstly, tachyzoites replicate by nuclear counts greater than binary in 1-3% in cell culture and this might be higher in alkaline media (needs to be evaluated). Secondly, we know that the centrosome of tachyzoites is uniquely bipartite with separate complexes controlling karyokinesis versus cytokinesis (budding). It is very easy to uncouple these mechanisms, which often leads to multinuclear replication. Abnormal multinuclear replication is one of the most common tachyzoite cell cycle mutant phenotypes. The images presented in this paper of large-scale nuclear reduplication are in most cases severely abnormal and are likely not viable--similar to cell cycle mutants. If the parasites are viable, the authors need to show that data. The centrin staining in this paper is unconvincing, and more importantly, the centrin complex controls budding not karyokinesis. The authors are using the wrong marker to follow the complex that controls nuclear replication (see Suvorova et al PLoS Biol).

Response: We thank the reviewer very much for the constructive suggestion. Indeed, abnormal multinuclear replication is one of the most common tachyzoite cell cycle mutant phenotypes. To address this point, we have employed different markers, including the inner centrosome core marker CEP250-L1, the kinetochore marker Nuf2, and the centromere marker TgChromo-1, related to karyokinesis to study nuclear division. Our results demonstrate that AP2XI-2-depleted parasites undergo endopolygony to produce mature merozoites. We would also like to indicate that the abnormal multinuclear structures cannot usually form daughter parasites with one nucleus per parasite. In contrast, the abnormal multinuclear AP2XI-2-depleted parasites can form typical daughter parasites with one nucleus per parasite, suggesting that these parasites have matured into merozoites.

Comment: The authors interpretation of IMC7 might need a reexamination. The inner membrane complex of daughter parasites is derived from both the mother and new synthesis, and thus, mature merozoites are positive for IMC7 from the mother. It is only the late schizont prior to budding completion that is IMC7 negative (look again at reference 27). More importantly, IMC7 is not just negative or positive here. IMC7 is extensively staining abnormal membranes (see Fig. 3c) so all bets are off about what this means developmentally, it is certainly not native IMC7 patterns.

Response: We have looked at Ref 27, in the later stages of schizogony (late schizont), it was possible to identify daughter IMC stained with IMC1, while IMC7 staining was limited to the periphery of the maternal parasite. Mature schizonts with fully formed merozoites were strongly stained with IMC1 but were completely negative for IMC7. Therefore, IMC7 extensively stains the abnormal membranes of the schizonts but not the mature merozoites in Fig 3c.

Comment: the authors had the right idea in using TEM to analyze parasite replication at higher resolution. However, three poor images with none showing daughter formation in their mutants falls far short of what was needed.

Response: To address this point, we have added more relevant information about bradyzoites, schizonts, and mature merozoites to the TEM figures.

Comment: Where are the IAA- staining images for tubulin and ARO protein in Fig. S2b?

Response: These have been added as requested.

Comment: It is not clear there is any value of the BFD1 knockout or knocking out XI-2 in RH experiments. Both the Pru parent and RH parent are developmentally compromised strains. The staining of XI-2-HA in the RH strain is non-nuclear and completely abnormal with or without IAA.

Response: In this experiment, we aimed to demonstrate that depletion of AP2XI-2 under alkaline conditions led the parasites to bypass the bradyzoite stage and switch directly to the merozoite, utilizing the BFD1 knockout. Using the RH strain, which has a rapid replication rate, we aimed to illustrate that the induction of merogony by alkaline medium may be a result of the parasites' slower growth. We have corrected the labeling of DNA and HA in Figure S3f.

Comment: Figure 7--other AP2 factors

The study of other AP2 factors in this paper contributes little to this paper and could be artifact on top of artifact. A more comprehensive study of XI-2 and XII-1 that includes evaluating the function of these two AP2s in developmentally competent strains is a better use of time and energy.

Response: We used AP2XI-2 or AP2XII-1 depleted parasites as an *in vitro* model to investigate the location and function of AP2s that are highly expressed in merozoite during merogony. The idea of using developmentally competent strains is worthwhile and should be pursued in future investigations and we have highlighted this important suggestion in the discussion. The part of the study relating to the identification of other AP2 factors revealed that AP2XI-2 is a part of an intricate regulatory network that controls merogony, providing the basis for further characterization of the role of these newly identified factors in the mechanisms underpinning merogony.

Responses to comments and suggestions of Reviewer #2

General comments:

The ultimate product of *Toxoplasma gondii* sexual development (the sporulated oocyst) accounts for roughly 50% of human infection. However, little is known at the molecular level about *Toxoplasma* pre-sexual development because this process is restricted to the cat gut and in vitro systems to study this stage are lacking. This manuscript details the role of two AP2 family transcription factors (AP2XI-2 & AP2XII-1) in preventing parasite sexual commitment by directing the HDAC3/MORC complex to transcriptionally silence gene expression. The work presented here closely mirrors a recent preprint (posted 01/2023, PMID: 36711883) by the Hakimi group. There is substantial overlap between this manuscript and the Hakimi preprint. Both works reach the same major conclusions: 1) AP2XI-2 & AP2XII-1 work in a complex to direct the HDAC3/MORC complex to transcriptionally silence gene expression, & 2) AP2XI-2 and/or AP2XII-1 knockdown drives merozoite formation. Whereas the Hakimi work is more developed on the genomics & proteomics side, the work reviewed here slightly extends beyond the Hakimi preprint by assessing the role of AP2XI-2/XII-1 accessory proteins to merozoite formation and validating the downstream expression of additional AP2 transcription factors that have roles yet to be defined. The manuscript tells a cohesive, complete, and intriguing story that would be of interest to broad readership interested in gene regulation and/or parasitology.

Response: We thank the reviewer very much for the favorable comments and constructive evaluation of our MS. Wherever relevant, we referred to the interesting recent findings made by Hakimi group and indicated how our new study provides new insights to expand the understanding of the scientific community about the role of the transcription factors AP2XI-2 & AP2XII-1 in regulating one of the most important but yet understudied aspects of the life cycle of *T. gondii*, the merogony process.

Comment: Fig 1H: the cyst burden is atypically low. What is the limit of detection in this assay? Was a PCR performed to measure parasite burden in the brain? Was serology performed to ensure infection in all mice? The methods section should state what the inoculum was in this section (not just the text body and figure legend). Data should be presented about weight loss over the time of acute infection.

Response: We thank the reviewer very much for constructive suggestions. The parasite cyst burden was determined based on the number of cysts detected in at least one cerebral hemisphere per mouse brain, by using an established method involving FITC-conjugated *Dolichos biflorus* lectin staining to count the parasite cysts. The theoretical limit number of detection is 1. All survived mice were confirmed to be infected by using serological testing. Mice infected by AP2XI-2 knockouts did not exhibit any toxoplasmosis symptoms, including weight loss, and we have revised this section and added a new figure (Fig. 1h in the revised

version) to present the body weight values measured over the course of 30 days post infection.

Comment: RNAseq analysis: states that bowtie2 was used for alignment which is atypical because it doesn't allow for spliced mapping. More importantly, the methods also state that differential gene expression analysis was performed with DESeq2. This data was not provided in supplement (should have log₂FC and stats associated with the call for each gene, not FPKM as provided). There are no volcano plots in the manuscript as presented (as mentioned in the methods) – just heatmaps for selected genes. It would be informative to show volcano plots though because it would allow the reader to assess to what degree genes are (dys)regulated under each condition. Preferably, genes presented in a volcano plot format could still be color-coded based on their canonical stage-specific expression. Sporadically throughout the manuscript, the RNAseq data is stated to show changes in protein expression as opposed to transcript abundance. Example: lines 300-1.

Response: We greatly appreciate the reviewer for the constructive suggestions, and we have incorporated comprehensive differential changes in mRNA expression (upregulated and downregulated genes) into the volcano plots, with genes presented in a color-coded format in a parasite stage-specific manner. In addition, we added the log₂ fold change data to the Supplementary Dataset 1. We have revised this section accordingly. We have also corrected the mention of protein expression to transcript abundance.

Comment: The deficiency in bradyzoite formation is detailed in Fig 2B & Fig 4B. What proportion of parasites are undergoing the merogony phenotype? What about in the deltaBFD1 line?

Response: We observed that more than 95% of the AP2XI-2-depleted or AP2XI-2-depleted deltaBFD1 parasites were undergoing merogony. We have added this data in the revised version.

Comment: Line 228-9: please substantiate the slow growth claim by assessing whether alkaline media does in fact reduce the growth rate independent of BFD1 expression in type your I/II parasite lines. (stated in ref 30 – lourido BFD1 paper)

Response: We have performed the requested study and confirmed that alkaline medium can reduce the growth of the deltaBFD1 strain, and we have now added this data to the revised MS.

Comment: To what degree to AP2XII1/AP2XI-2 and MORC peaks overlap? What proportion of MORC-resident genes are XII-1/XI-2 dependent? Does MORC occupancy decrease at these genes upon XII1/XI2 depletion?

Response: We thank the reviewer very much for constructive comments. We observed that most genes bound by AP2XI-2 and AP2XII-1 showed MORC occupancy, with a clear preference for genes bound by all three proteins. Furthermore, MORC occupancy at these

genes decreased upon depletion of AP2XI-2.

Comment: Line 310-2: if both AP2XII-1 and XI-2 bind each other (co-IP experiment), probably acting as a heterodimer, what model is proposed where one works upstream of another? Are there peaks that are uniquely APXII-1 or AP2XII-2 from the CUT&Tag to support a model where these factors could work independently? For example could they form a mixture of homo- and heterodimers that could explain their differences?

Response: We thank the reviewer very much for constructive suggestions. We observed that most genes bound by AP2XI-2 also exhibited occupancy by AP2XII-1. The CUT&Tag data did not provide evidence to support a model by which these two factors would work independently. We have revised the MS accordingly.

Comment: Fig 5A: the last column shows egressed merozoites. Can they reinvade? And what happens to do them under the second growth cycle and beyond?

Response: The mature merozoites could not re-invade the host cell even under normal medium conditions. We have added this data in the revised MS.

Comment: Fig 6: since statements are given in the text about the relative switching efficiency under neutral and alkaline culture conditions, please categorize and quantify in the figure. This is relevant because the statistical term “significantly increased” is used on line 307. Without this information, it is hard to ascertain to what degree the culture conditions matter in the double knockdown. Same comment applies throughout the paragraphs from lines 306-323 since quantitative comparisons are being made.

Response: We thank the reviewer very much for constructive suggestions, and we have added the quantitative data, together with the quantitative statistical analysis, based on the endopolygeny in Fig. 6c.

Comment: Fig 7A: are each of these AP2s turned on only by XI-2 knockdown? Please show IFA to see whether they are turned in under normal neutral or alkaline (without IAA) conditions. Also, which (if any) of these have been shown to complex with MORC – would give indication about whether they are transcriptional activators or repressors.

Response: We thank the reviewer very much for constructive suggestions, and we have added this data in the Supplementary figure 8a. In addition, these factors were not bound to the MORC complex.

MINOR:

Point 1: Line 122: Suggest removing the term “two-color assay” assay since it is not mentioned by that name in the Materials and Methods section and the term is not informative to a general readership audience. Alternatively, modify M&M accordingly.

Response: We have modified M&M accordingly.

Point 2: Lines 141-52: given the generalist readership of this journal, explain the importance of the IMC1 staining as marking the mother in this paragraph and how that contributes to the conclusion that the parasites have ‘multinucleated morphology’ as mentioned in the text.

Response: We have clarified the reason for IMC1 staining as suggested.

Point 3: Fig S2D: what is the red stain for this panel? If IMC1, it would be helpful to recolor the figure to mirror S2C.

Response: Changes were implemented as suggested.

Point 4: Lines 167-8: soften the language about these genes being tachyzoite-specific (SAG1 is tachyzoite specific; GRA1 is in bradyzoites PMID: 18006371; MIC2 is lowered but not gone in bradyzoites PMID: 34860156; not sure about ROP1). What about other canonical tachyzoite markers (like ENO2)? Suggest to reword saying that their transcripts have their peak expression in tachyzoites if this is the case.

Response: Revised accordingly.

Point 5: Fig S2B typo (IMC17A->MIC17A)

Response: Corrected accordingly.

Point 6: Fig S2C: please provide labels showing key features described in the text on the micrographs.

Response: The labels showing key features described in the text on the micrographs were added as suggested.

Point 7: Fig S2D: what is the red stain for this panel?

Response: We have added IMC1 accordingly.

Point 8: Line 211: typo “was no observed”

Response: Revised accordingly.

Point 9: Fig S3F: panels are mislabeled (HA / DNA).

Response: Revised accordingly.

Point 10: Line 254: typo were->was

Response: Revised accordingly.

Point 11: Line 291: typo nucleuses->nuclei

Response: Revised accordingly.

Point 12: Fig 6: each panel is very small and hard to see what is going on without really zooming in.

Response: We have split Figure 6 into two separate images to enhance the visibility.

Point 13: Fig 7A: suggest adding a label to the figure to indicate the categories mentioned in the text to allow for quick referral (e.g. IV-3, IV-2, VIIa-1 are present in mature merozoites).

Response: We have annotated the labels of the figures to the left side of the corresponding images to indicate the 9 different types of the transcriptional factors mentioned in the text.

Point 14: Line 388: change “unable to regulate bradyzoite genes” to “unable to induce bradyzoite genes”

Response: Revised accordingly.

Point 15: Line 437: typo w,hereas

Response: Revised accordingly.

Point 16: Line 600: what is the elution buffer?

Response: Revised accordingly.

Point 17: Reference used for the MORC ChIPseq data should be provided in the appropriate section of the methods and in the body of the text.

Response: Added accordingly.

Responses to comments and suggestions of Reviewer #3

General comments:

In this manuscript, Wang and colleagues identify two putative transcription factors from the eukaryotic pathogen *Toxoplasma gondii* that play a critical role in regulating stage-specific gene expression in the parasite. They demonstrate that AP2XII-1 and AP2XI-2 are expressed in the tachyzoite and bradyzoite stages and repress genes that are normally expressed during merogony and in the mature merozoite. Knockdown of either (or both) of these regulators results in upregulation of merozoite-specific genes with concurrent downregulation of tachyzoite genes and is associated with changes in parasite morphology that is consistent with merogony and mature merozoites. This is further supported with a large number of inducible knockdown lines of genes encoding proteins that interact with AP2XII-1 and AP2XI-2. Importantly, this work demonstrates the feasibility of in vitro production and study of *Toxoplasma* merozoites and will be of great interest to the Apicomplexan field.

The study is complementary to a recent preprint from Antunes et al (<https://doi.org/10.1101/2023.01.16.524187>) and the conclusions from both groups are aligned. Although there is a degree of overlap between the central hypotheses of the two

manuscripts, an important distinction is that Wang et al have performed their analyses in a Type II Pru parasite background (compared to the Type I RH background in Antunes et al). Significantly, Wang et al describe AP2 knockdown in the context of alkaline stress which appears to be a strong trigger of merogony in the absence of either XI-2 or XII-1.

The experiments have been performed rigorously with sufficiently detailed methods in most cases.

Response: We thank the reviewer very much for the favorable comments and constructive suggestions on our MS. We are pleased to see the thoughtful comment of the reviewer considering the results obtained in our study be of great interest to the Apicomplexan field and aligned with results obtained in a recent preprint. We are also pleased to see the key and novel data obtained in our study are well recognized/acknowledged by the reviewer.

Minor comments and suggestions:

Comments: The change in morphology to “significantly thinner” or “multinucleated” is used as a marker for merogony and merozoite formation throughout the paper, but all the images that are presented are zoomed out, and in some cases (eg Fig 2a) the DAPI channel is unfocused and blurry, making it difficult to clearly define these different morphologies. It would be helpful to include a few enlarged inset images of just a few parasites to clearly demonstrate these two morphologies and allow the reader to compare to previously published images of enteroepithelial stages.

Response: We thank the reviewer very much for the constructive suggestion. Because the multinucleated schizonts exhibit several morphologies, we have included several examples in figure 2a and other figures to show these morphological abnormalities. We have also improved the figure resolution. For annotations, we specifically labeled Figure 3c, as Figures 1 and 2 did not introduce the abnormal parasites as schizonts and merozoites in the MS. Additionally, in Fig S2d, there are representative transmission electron microscopic micrographs of AP2XI-2 depleted parasites, showing AP2XI-2 depleted parasites undergoing merogony and produce mature merozoites.

Comments: Most of the figures containing imaging data include 3 or more examples of the different morphologies that were observed (figs. 2a, 3c, 5 and 6). The authors might consider streamlining this data a little to help the reader, and include just one example each of the “thinner” and “multinucleated” morphologies. Furthermore, annotating the images with arrows or labels of putative schizonts/merozoites would help with reader comprehension.

Response: We thank the reviewer very much for the constructive suggestions. Because the multinucleated schizonts exhibit several morphologies, we have included several examples in figure 2a and other figures to show these morphological abnormalities. We have also improved the figure resolution. For annotations, we specifically labeled Figure 3c, as Figures 1 and 2 did not introduce the abnormal parasites as schizonts and merozoites in the MS. Additionally, in Fig S2d, there are representative transmission electron microscopic

micrographs of AP2XI-2 depleted parasites, showing AP2XI-2 depleted parasites undergoing merogony and produce mature merozoites.

Comments: How were “bradyzoite-specific” and “merozoite-specific” genes defined in Figure 2 and 3? What were the criteria used to select these lists of genes?

Response: These are the genes that are typically upregulated/highly expressed in specific *T. gondii* stage. Hence, we have changed “bradyzoite-specific” and “merozoite-specific” to “bradyzoite highly expressed” and “merozoite highly expressed”. The gene cluster shown in Figure 2 and Figure 3 was obtained from Supplementary Table 4 in a previous study (PMID: 32094587).

Comments: Figure 2b. How was “high, middle and low” DBL staining defined? This should be explained in the methods.

Response: We have explained the stratification of the DBL staining in the methods. We have also added image references for the DBL staining criteria as requested.

Comments: Line 291: nucleuses should be nuclei

Response: Revised accordingly.

Comments: Line 844, Fig 1c legend: (c) should be (b)

Response: Revised accordingly.

We have also carefully proofread the entire MS and corrected other minor mistakes and typo-errors.

Sincerely yours,

Xing
Xing-Quan Zhu, BVSc, MVSc, PhD
Professor & Head, Laboratory of Parasitic Diseases,
College of Veterinary Medicine,
Shanxi Agricultural University,
Taigu, Shanxi Province 030801,
People’s Republic of China
Email: xingquanzhu1@hotmail.com

REVIEWERS' COMMENTS

Reviewer #1 (Remarks to the Author):

The manuscript entitled "The transcription factor AP2XI-2 is a key negative regulator" has been improved. The authors were responsive to most of the reviewer comments. In responding to the reviewers the authors indicated that only 10-15% of schizonts were producing merozoites. I don't believe this data was included in the paper, but is important for the reader to understand the limits of this model. Likewise, the authors explain this result as the reason only half the known merozoite mRNAs were induced including the weak expression of the dominant merozoite-specific AP2. Add to this the inability of the merozoites to re-invade HFF cells points to an incomplete model. The concept of incompleteness of the experimental model needs to be more than a private conversation with the reviewers.

Include the 10-15% result and add a sentence or two about the model limitations to produce functional merozoites in the discussion and the paper would be acceptable to this reviewer.

Reviewer #2 (Remarks to the Author):

This resubmission has addressed all issues raised from the initial review. A couple minor points:

- line 339: typo "guiding this" -> "guiding these"

- The revised discussion includes a reference to the preprint by the Hakimi group that significantly overlaps this work. Another work from the Shen group with significant overlap and reaching similar conclusions was published in September and should and should be cited in a similar context [PMID: 37750704]. That the topic has attracted interest from several groups speaks to the timeliness and impact of the work.

Responses to reviewers' comments

Reviewer #1 (Remarks to the Author):

The manuscript entitled "The transcription factor AP2XI-2 is a key negative regulator" has been improved. The authors were responsive to most of the reviewer comments. In responding to the reviewers the authors indicated that only 10-15% of schizonts were producing merozoites. I don't believe this data was included in the paper, but is important for the reader to understand the limits of this model. Likewise, the authors explain this result as the reason only half the known merozoite mRNAs were induced including the weak expression of the dominant merozoite-specific AP2. Add to this the inability of the merozoites to re-invade HFF cells points to an incomplete model. The concept of incompleteness of the experimental model needs to be more than a private conversation with the reviewers.

Include the 10-15% result and add a sentence or two about the model limitations to produce functional merozoites in the discussion, and the paper would be acceptable to this reviewer.

Response: We thank the reviewer very much for the favorable comments and constructive suggestions on our revised manuscript. We have addressed the limitations related to producing fully functional merozoites using the model presented in the revised manuscript as suggested. We have included the information showing that approximately half of the known merozoite mRNAs were induced after the depletion of AP2XI-2 in the discussion, and provided a suggestion for future investigation in the conclusion.

Reviewer #2 (Remarks to the Author):

Comment: This resubmission has addressed all issues raised from the initial review. A couple minor points:

Point 1: line 339: typo "guiding this" -> "guiding these"

Response: Revised accordingly.

Point 2: The revised discussion includes a reference to the preprint by the Hakimi group that significantly overlaps this work. Another work from the Shen group with significant overlap and reaching similar conclusions was published in September and should be cited in a similar context [PMID: 37750704]. That the topic has attracted interest from several groups speaks to the timeliness and impact of the work.

Response: We thank the reviewer very much for favorable comments on the timeliness and impact of the research presented in our revised manuscript and recently published articles. We have cited the publication authored by the Shen group as suggested.